# The PARTNER trial of neoadjuvant olaparib with chemotherapy in triple-negative breast cancer

Jean E. Abraham[1,2 ✉], Karen Pinilla[1,2], Alimu Dayimu[3], Louise Grybowicz[4], Nikolaos Demiris[5], Caron Harvey[4], Lynsey M. Drewett[6], Rebecca Lucey[1,2], Alexander Fulton[1,2], Anne N. Roberts[4], Joanna R. Worley[1,2], Anita Chhabra[7], Wendi Qian[8], Anne-Laure Vallier[4], Richard M. Hardy[4], Steve Chan[9], Tamas Hickish[10], Devashish Tripathi[11,12], Ramachandran Venkitaraman[13], Mojca Persic[14], Shahzeena Aslam[15], Daniel Glassman[16], Sanjay Raj[17,18,19], Annabel Borley[20], Jeremy P. Braybrooke[21], Stephanie Sutherland[22], Emma Staples[23], Lucy C. Scott[24], Mark Davies[25], Cheryl A. Palmer[26], Margaret Moody[27], Mark J. Churn[28,29,30], Jacqueline C. Newby[31], Mukesh B. Mukesh[32], Amitabha Chakrabarti[33], Rebecca R. Roylance[34], Philip C. Schouten[35], Nicola C. Levitt[36], Karen McAdam[37], Anne C. Armstrong[38], Ellen R. Copson[39], Emma McMurtry[40], Marc Tischkowitz[41], Elena Provenzano[35] & Helena M. Earl[1,2]

PARTNER is a prospective, phase II–III, randomized controlled clinical trial that recruited patients with triple-negative breast cancer[1,2], who were germline *BRCA1* and *BRCA2* wild type[3]. Here we report the results of the trial. Patients (*n* = 559) were randomized on a 1:1 basis to receive neoadjuvant carboplatin–paclitaxel with or without 150 mg olaparib twice daily, on days 3 to 14, of each of four cycles (gap schedule olaparib, research arm) followed by three cycles of anthracycline-based chemotherapy before surgery. The primary end point was pathologic complete response (pCR)[4], and secondary end points included event-free survival (EFS) and overall survival (OS)[5]. pCR was achieved in 51% of patients in the research arm and 52% in the control arm (*P* = 0.753). Estimated EFS at 36 months in the research and control arms was 80% and 79% (log-rank *P* > 0.9), respectively; OS was 90% and 87.2% (log-rank *P* = 0.8), respectively. In patients with pCR, estimated EFS at 36 months was 90%, and in those with non-pCR it was 70% (log-rank *P* < 0.001), and OS was 96% and 83% (log-rank *P* < 0.001), respectively. Neoadjuvant olaparib did not improve pCR rates, EFS or OS when added to carboplatin–paclitaxel and anthracycline-based chemotherapy in patients with triple-negative breast cancer who were germline *BRCA1* and *BRCA2* wild type. ClinicalTrials.gov ID: NCT03150576.

Olaparib, the first poly(ADP-ribose) polymerase (PARP) inhibitor to be developed, is effective in treating women with breast cancer who have germline pathogenic variants in *BRCA1* and/or *BRCA2* (gBRCAm), both in the metastatic[6,7] and the adjuvant setting[8]. The PARTNER trial tested olaparib in the neoadjuvant setting in two cohorts. One cohort consisted of the patients with gBRCAm who had early breast cancer and the report of that cohort is in progress (J.E.A. et al., manuscript in preparation). The other cohort (reported here) consisted of patients with triple-negative breast cancer (TNBC) who were germline *BRCA1* and *BRCA2* wild type (gBRCAwt)). In addition, all tumours had a basal-like phenotype as defined by immunohistochemistry (Methods). The standard of care for many years for TNBC had been anthracycline- and taxane-based chemotherapy[9]. However, there was emerging evidence from neoadjuvant trials (now published) that carboplatin is a useful addition to this standard treatment[10,11]. In our trial, olaparib was given 48 h after carboplatin–paclitaxel (gap schedule) for four cycles in the

neoadjuvant setting and was followed by anthracycline-based chemotherapy before surgery.

TNBCs in patients who are gBRCAwt frequently exhibit homologous recombination deficiency, and widespread genomic instability[1] similar to that seen in breast cancers in patients with gBRCAm. Defects in homologous recombination repair can occur through numerous mechanisms including the loss of *BRCA1* and *BRCA2* function within the breast cancer, thus resulting in a gBRCAm-like phenotype[12], which could potentially be treated with PARP inhibitors. 'Genomic scars', typically found in gBRCAm-related tumours, have also been identified in tumours that are gBRCAwt[13]. Typical rearrangement signatures with high numbers of tandem duplications have been linked to a subgroup of TNBC(gBRCAwt) with a homologous recombination deficiency profile[14]. This suggests that TNBC(gBRCAwt) includes a targetable molecular group outside the gBRCA population that could also benefit from PARP inhibition. In homologous-recombination-deficient cells, PARP inhibition results in

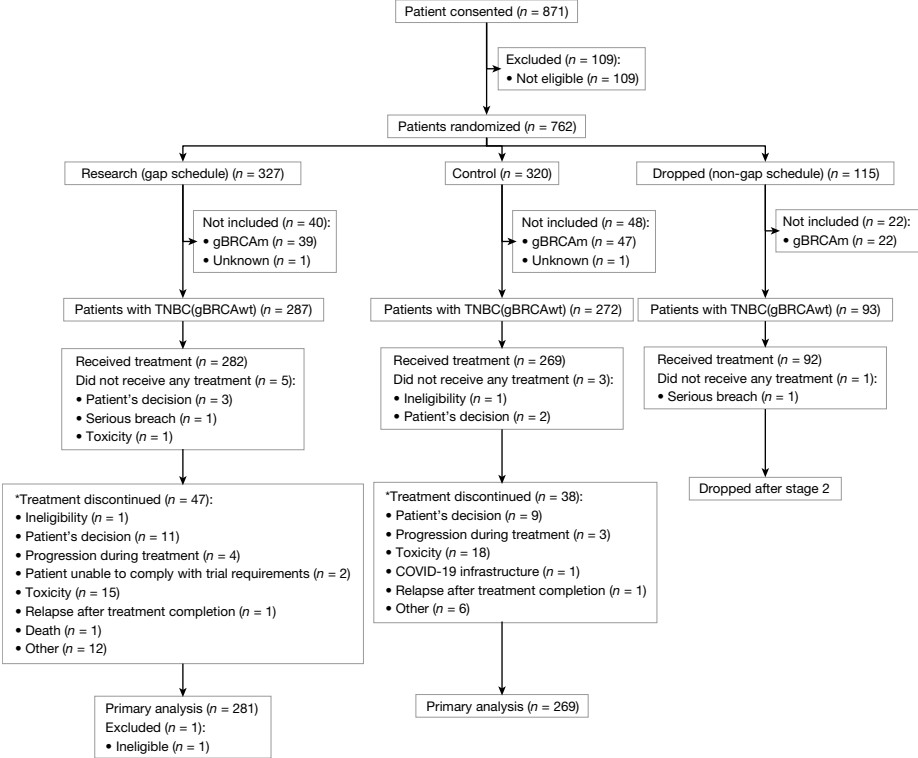

Patient consented (*n* = 871)

Excluded (*n* = 109):
• Not eligible (*n* = 109)

Patients randomized (*n* = 762)

Research (gap schedule) (*n* = 327) | Control (*n* = 320) | Dropped (non-gap schedule) (*n* = 115)

Not included (*n* = 40):
• gBRCAm (*n* = 39)
• Unknown (*n* = 1)

Not included (*n* = 48):
• gBRCAm (*n* = 47)
• Unknown (*n* = 1)

Not included (*n* = 22):
• gBRCAm (*n* = 22)

Patients with TNBC(gBRCAwt) (*n* = 287) | Patients with TNBC(gBRCAwt) (*n* = 272) | Patients with TNBC(gBRCAwt) (*n* = 93)

Received treatment (*n* = 282)
Did not receive any treatment (*n* = 5):
• Patient's decision (*n* = 3)
• Serious breach (*n* = 1)
• Toxicity (*n* = 1)

Received treatment (*n* = 269)
Did not receive any treatment (*n* = 3):
• Ineligibility (*n* = 1)
• Patient's decision (*n* = 2)

Received treatment (*n* = 92)
Did not receive any treatment (*n* = 1):
• Serious breach (*n* = 1)

Dropped after stage 2

*Treatment discontinued (*n* = 47):
• Ineligibility (*n* = 1)
• Patient's decision (*n* = 11)
• Progression during treatment (*n* = 4)
• Patient unable to comply with trial requirements (*n* = 2)
• Toxicity (*n* = 15)
• Relapse after treatment completion (*n* = 1)
• Death (*n* = 1)
• Other (*n* = 12)

*Treatment discontinued (*n* = 38):
• Patient's decision (*n* = 9)
• Progression during treatment (*n* = 3)
• Toxicity (*n* = 18)
• COVID-19 infrastructure (*n* = 1)
• Relapse after treatment completion (*n* = 1)
• Other (*n* = 6)

Primary analysis (*n* = 281)
Excluded (*n* = 1):
• Ineligible (*n* = 1)

Primary analysis (*n* = 269)

**Fig. 1 | Trial CONSORT diagram.** *The first main reason for treatment discontinuation was reported.

synthetic lethality by preventing repair of single-strand breaks, which leads to problems downstream with double-strand repair[15,16]. PARP inhibitors therefore could work in synergy with DNA-damaging agents such as platinums, which cause both single- and double-strand breaks. In high-grade serous ovarian cancer, widespread adoption of PARP inhibitors in gBRCAwt tumours has already occurred and has been supported by evidence of their activity in those cancers that demonstrate homologous recombination deficiency[17,18].

The PARTNER trial used olaparib in the experimental group as an addition to carboplatin–paclitaxel followed by anthracycline-based chemotherapy. The first stage of the trial examined the safety of the combination and the second stage investigated optimal scheduling of olaparib in combination with platinum chemotherapy, which has never been established. The combination had previously been tested in patients with relapsed ovarian cancer[19], but the dose and schedule used required reduced doses of carboplatin, and the combination resulted in response rates similar to, rather than superior to, those for carboplatin–paclitaxel alone. Therefore, after these results, olaparib has been scheduled after completion of chemotherapy, when full and continuous doses are used[8]. The third stage of the PARTNER trial investigated the efficacy of the same chemotherapy and olaparib in the 48-h gap schedule research arm (Extended Data Fig. 1). This paper details the results of the PARTNER trial, a prospective, randomized controlled trial in the TNBC(gBRCAwt) cohort.

## Patients and treatment

From September 2016 to December 2021, 559 patients with TNBC(gBRCAwt) (287 research arm (gap schedule); 272 control arm) were randomized at 29 UK centres (CONSORT diagram, Fig. 1). Recruitment was extended by around 6 months to decrease the risk of losing participants due to the COVID-19 pandemic, which had considerably slowed recruitment during 2020.

The data cutoff for analysis was 30 November 2023 with a median follow-up of 38 months. Five patients (3 opt-out, 1 breach, 1 toxicity)

in the research arm (gap schedule) and three patients (1 ineligible, 2 opt-out) in the control arm did not receive any treatment after randomization. In the research arm (gap schedule), one patient was found to be ineligible after receiving treatment. The resulting modified intention-to-treat population consisted of 550 patients. The demographics and pretreatment disease characteristics were well balanced between the two arms (Table 1). The median patient age was 49 (range 23 to 71) years; 95.1% of patients had a tumour diameter of less than 50 mm; 22.5% had a tumour-infiltrating lymphocyte (TIL) score ≥ 60%; 94.8% had an Eastern Cooperative Oncology Group performance status of 0; and 36.5% had prior oophorectomy or were post-menopausal. In the research arm (gap schedule), 88.2% received at least 80% of the planned olaparib with a median dose intensity of 1,170 mg per week (Extended Data Table 1). There were no differences between the research (gap schedule) and control arms for patients receiving at least 80% of the planned carboplatin (95%) and paclitaxel doses (99%) (Extended Data Table 1). Surgery was carried out after the treatment was completed in 98.2% of the patients (276 research (gap schedule); 264 control). In patients who had surgery, 73.9% had breast-conserving wide local excision and 61.5% had sentinel node biopsy. A total of 447 (81.3%) patients received local radiotherapy according to centre protocols, after completion of trial treatment and surgery (Table 1).

## Efficacy

A total of 543 patients had pathological response data at surgery available, of which 141/276 (51.1%) in the research arm (gap schedule) and 140/267 (52.4%) in the control arm had a pCR with a difference of −1.3% (95% confidence interval (CI) −9.7% to 7.0%, *P* value = 0.753; Fig. 2a). The result of no significant difference was consistent in all pre-specified subgroups and after imputing missing data over a range of assumptions (Extended Data Table 2 and Extended Data Fig. 2). The proportion of patients with pCR was higher for those with tumours with TILs ≥ 60% (65%) compared to those with TILs < 60% (47.9%) with

**Table 1 | Baseline characteristics and surgery carried out for the patients with at least one dose of treatment**

| Variable | Research (gap schedule) (*n*=281) | Control (*n*=269) | Total (*n*=550) |
|---|---|---|---|
| **Median age (range)[a]** | 49.6 (23.9, 70.9) | 48.4 (23.2, 71.0) | 49.1 (23.2, 71.0) |
| **Ethnicity, *n* (%)** | | | |
| White | 228 (81.1%) | 217 (80.7%) | 445 (80.9%) |
| Mixed | 2 (0.7%) | 2 (0.7%) | 4 (0.7%) |
| Asian or Asian British | 4 (1.4%) | 7 (2.6%) | 11 (2.0%) |
| Black or Black British | 9 (3.2%) | 4 (1.5%) | 13 (2.4%) |
| Unknown | 38 (13.5%) | 39 (14.5%) | 77 (14.0%) |
| **Tumour size, *n* (%)** | | | |
| ≤50 mm | 265 (94.3%) | 258 (95.9%) | 523 (95.1%) |
| >50 mm | 16 (5.7%) | 11 (4.1%) | 27 (4.9%) |
| **Axillary lymph node involvement at diagnosis by biopsy and/or imaging, *n* (%)** | | | |
| No | 188 (66.9%) | 188 (69.9%) | 376 (68.4%) |
| Yes | 93 (33.1%) | 81 (30.1%) | 174 (31.6%) |
| **TIL score, *n* (%)** | | | |
| <60% | 215 (76.5%) | 211 (78.4%) | 426 (77.5%) |
| ≥60% | 66 (23.5%) | 58 (21.6%) | 124 (22.5%) |
| **Eastern Cooperative Oncology Group performance status, *n* (%)** | | | |
| 0 | 260 (92.5%) | 258 (95.9%) | 518 (94.2%) |
| 1 | 21 (7.5%) | 11 (4.1%) | 32 (5.8%) |
| **HER2 immunohistochemisty status, *n* (%)** | | | |
| 0 | 233 (82.9%) | 220 (81.8%) | 453 (82.4%) |
| 1 | 28 (10.0%) | 24 (8.9%) | 52 (9.5%) |
| 2 | 20 (7.1%) | 25 (9.3%) | 45 (8.2%) |
| **Post-menopausal (oophorectomy before diagnosis or natural menopause)** | | | |
| Yes | 95 (35.4%) | 96 (37.5%) | 191 (36.5%) |
| No | 173 (64.6%) | 160 (62.5%) | 333 (63.5%) |
| **Surgery after neoadjuvant treatment[b], *n* (%)** | 276 (98.2%) | 264 (98.1%) | 540 (98.2%) |
| **Breast surgery for the protocol-treated breast cancer[b], *n* (%)** | | | |
| Breast-conserving wide local excision | 200 (72.5%) | 199 (75.4%) | 399 (73.9%) |
| Mastectomy | 79 (28.6%) | 67 (25.4%) | 146 (26.5%) |
| Reconstruction | 24 (8.7%) | 16 (6.1%) | 40 (7.4%) |
| **Axillary surgery for the protocol-treated breast cancer[c], *n* (%)** | | | |
| Sentinel node biopsy | 166 (60.1%) | 166 (62.9%) | 332 (61.5%) |
| Axillary clearance | 88 (31.9%) | 78 (29.5%) | 166 (30.7%) |
| Axillary sampling | 42 (15.2%) | 43 (16.3%) | 85 (15.7%) |
| **Radiotherapy (local) after completion of protocol treatment** | 231 (82.2%) | 216 (80.3%) | 447 (81.3%) |

[a]Maximum age of 71 was due to rounding.
[b]A total of ten patients (five in each group) had missing surgery information. The denominators are the number of patients in each group.
[c]Each patient may have several surgeries; denominators are the number of patients who had a surgery.

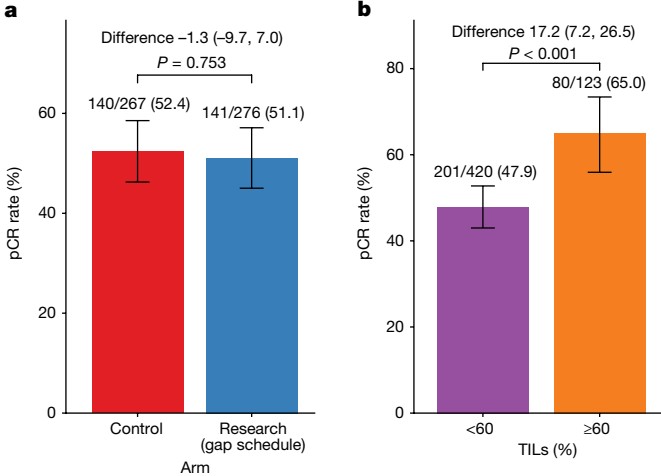

**Fig. 2 | pCR rate by treatment arm and TIL group. a,b**, pCR rate by treatment arm (**a**) and TIL group (≥60% versus <60%) (**b**). Data were analysed from a total of 276 patients in the research (gap schedule) and 267 control arms. Error bars, 95% CI of the proportion based on exact method. The statistical test was based on the two-tailed chi-squared test.

75.2 to 85.5) in the research arm (gap schedule) and 79.1% (95% CI 73.9 to 84.7) in the control arm, with a median EFS not reached in either (Extended Data Table 4).

A total of 63 patients (31 research (gap schedule); 32 control) died, with an HR of 0.9 (95% CI 0.6 to 1.5, *P* = 0.8; Fig. 3b). The estimated OS rate at 36 months was 90.3% (95% CI 86.5 to 94.2) in the research arm (gap schedule) with a median OS not reached, and 87.2% (95% CI 82.8 to 91.9) in the control arm with a median OS of 74.7 months (Extended Data Table 4).

A total of 77 patients (42 research (gap schedule); 35 control) had a distant relapse or died with an HR of 1.1 (95% CI 0.7 to 1.7, *P* = 0.632; Fig. 3c). The estimated distant disease-free survival (DDFS) rate at 36 months was 85.5% (95% CI 81.2 to 90.1) in the research arm (gap schedule) and 86.2% (95% CI 81.7 to 90.9) in the control arm, with a median DDFS not reached in either (Extended Data Table 4).

Similarly, no difference was observed in relapse-free survival (HR = 1.0, 95% CI 0.7 to 1.4, *P* = 0.896), local recurrence-free survival (HR = 0.9, 95% CI 0.6 to 1.5, *P* = 0.750), time to second cancer (HR = 0.5, 95% CI 0.2 to 1.2, *P* = 0.126) or breast cancer-specific survival (HR = 1.0, 95% CI 0.6 to 1.6, *P* = 0.902; Extended Data Fig. 3a–d). Neither arm reached a median on these time-to-event outcomes, except the control arm for OS (74.7 months) and breast cancer-specific survival (74.7 months; Extended Data Table 4).

An exploratory analysis was carried out including the 92 patients randomized into the dropped arm (non-gap schedule), compared with patients at stage 3, control and research (gap schedule) arms (Extended Data Fig. 4), and with patients at stage 2, control and research (gap schedule) arms (Extended Data Fig. 5). There were no significant differences for estimated EFS, OS or DDFS in any comparisons.

Kaplan–Meier curves of EFS and OS by pathological response and treatment group are presented in Fig. 4. The estimated 36-month EFS rate was 90.4% (95% CI 86.4 to 94.5) in the patients with a pCR as compared with 70% (95% CI 64.2 to 76.2) in those with a non-pCR (HR = 0.3, 95% CI 0.2 to 0.4; *P* < 0.001). Similarly, more deaths were observed in patients with a non-pCR (*P* < 0.001). The estimated 36-month OS rate was 95.7% (95% CI 93.0 to 98.5) in the patients with a pCR as compared with 83% (95% CI 78 to 88.2) in those with a non-pCR (HR = 0.2, 95% CI 0.1 to 0.3; *P* < 0.001). Fewer events and deaths were observed in patients with a pCR compared to those with a non-pCR, regardless of the treatment received (Fig. 4b,d).

a difference of 17.2% (95% CI 7.2% to 26.5%, *P* value < 0.001; Fig. 2b). There were no significant differences in pCR rate in each TIL group between the research (gap schedule) and control arms (Extended Data Fig. 2).

A total of 110 patients (57 research (gap schedule); 53 control) had an event or died with a hazard ratio (HR) of 0.9 (95% CI 0.6 to 1.4, *P* = 0.781; Fig. 3a). A detailed breakdown of the event types is shown in Extended Data Table 3. The estimated EFS rate at 36 months was 80.2% (95% CI

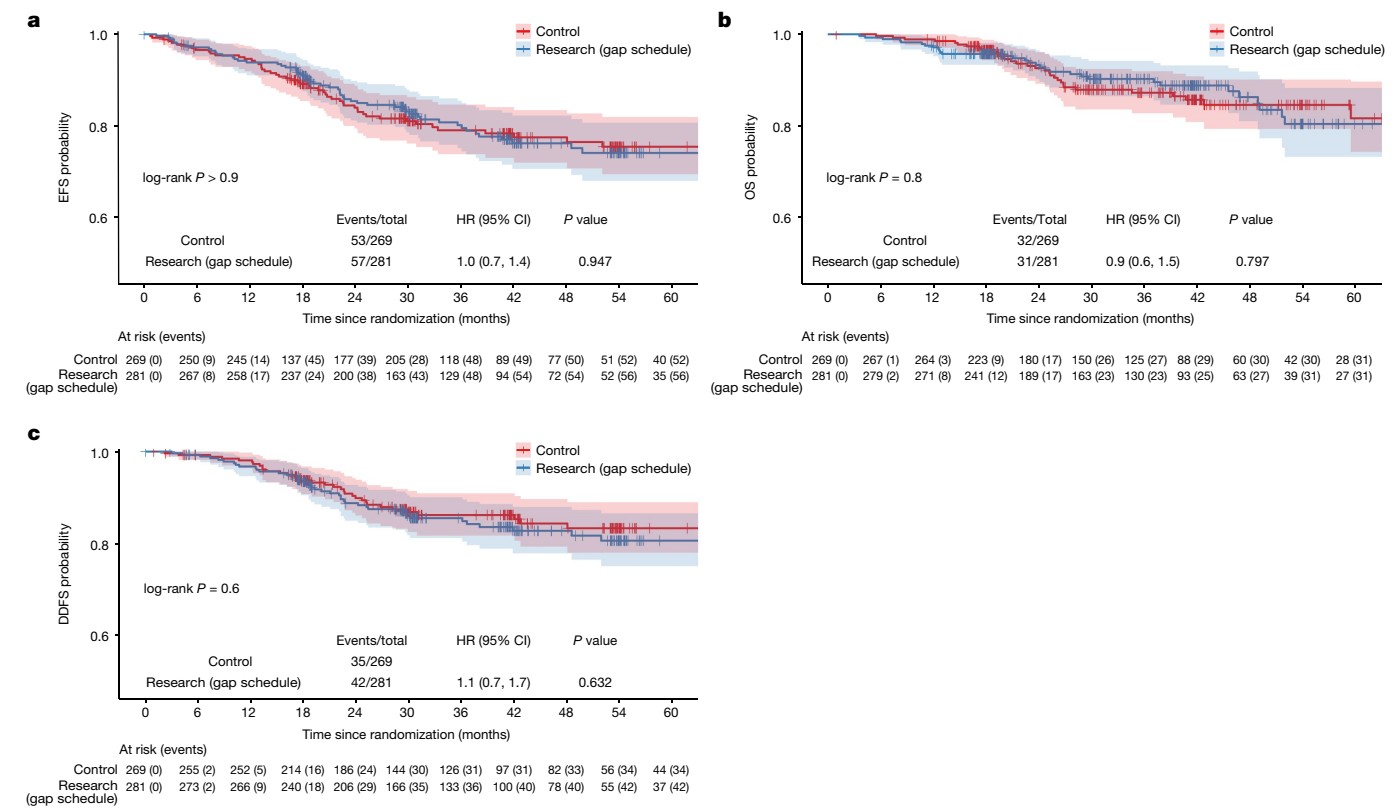

**Fig. 3 | Kaplan–Meier curves of key survival end points. a–c**, EFS (**a**), OS (**b**) and DDFS (**c**) in patients in the modified intention-to-treat group by treatment arm.

## Safety and toxicity

A total of 551 (282 research (gap schedule); 269 control) patients were evaluated for safety. A total of 47 (16.6%) patients in the research arm (gap schedule) and 38 (14.1%) in the control arm stopped treatment early with toxicity being the most common reason (21 research (gap schedule); 22 control group). Of patients who discontinued trial treatment early, eight in the research (gap schedule) arm and three in the control arm had further neoadjuvant treatment outside the protocol.

Patients in the research arm experienced slightly more adverse events (AEs) of grade ≥3 than those in the control arm (64.2% versus 58.7%; Table 2). The number of serious AEs (SAEs) related to carboplatin was slightly higher in the research arm (60 (21.3%)) than in the control arm (49 (18.2%)) and the number of SAEs related to paclitaxel was also slightly higher in the research arm (63 (22.3%)) than in the control arm (47 (17.5%)). A total of 45 (16.0%) patients in the research arm had an SAE related to olaparib. During the whole treatment period, the number of patients who had a transfusion was higher in the research arm (145 (51.4%)) than in the control arm (82 (30.5%)). A summary of the worst AE grade ≥3 experienced per patient in at least 1% of patients is shown in Extended Data Table 5, and the only AE that is significantly worse in the research arm compared with the control arm is neutropenia without associated fever (research arm, 95 (33.7%); control arm, 50 (18.6%); $P = 0.002$). More patients in the research arm than patients in the control arm had a missed or modified dose of carboplatin (20.2% versus 9.7%, respectively) or a missed or modified dose of paclitaxel (52.1% versus 35.7%, respectively) due to toxicity (see Extended Data Table 6 for full details).

## Quality of life

A total of 522 patients (268 research (gap schedule); 254 control) consented to the quality of life sub-study. Please refer to Methods, Study procedures for explanation of the health-related quality of life (HRQOL) measures. The mean EQ-5D-5L (EuroQol 5 Dimension 5 Level) visual analogue scale score and Functional Assessment of Cancer Therapy–Breast (FACT-B) total score were consistently slightly higher (slightly better HRQOL) for the control arm compared with the research arm (gap schedule) throughout the study period (Extended Data Fig. 6a,b). However, the only data showing a significantly worse HRQOL for patients in the research (gap schedule) arm compared with those in the control arm was at the 3-month time point for the FACT-B total score (Extended Data Fig. 6c). A decrease was observed in both the control and research (gap schedule) arms at 4 and 6 months in physical well-being, social and family well-being, functional well-being, breast cancer subscale, FACT-B trial outcome index and FACT-B total score from the baseline group, with slightly larger decreases in the research arm (gap schedule), which were non-significant.

## Discussion

This neoadjuvant trial tested low-dose, intermittent olaparib, 150 mg twice daily by mouth for 12 days, starting on day 3 until day 14, every 3 weeks for four cycles, concurrently with four cycles of carboplatin–paclitaxel, in patients with TNBC(gBRCAwt). This treatment was followed by three cycles of anthracycline-based chemotherapy. There was no improvement in either estimated EFS and OS at 36 months or pCR rate at surgery from the addition of olaparib. Our original hypothesis was that low-dose olaparib would work in synergy with platinum-based chemotherapy to enhance lethal damage to cancer cells and increase the rate of pCR. This hypothesis has been disproved in the TNBC(gBRCAwt) cohort. However, results for the gBRCAm cohort from the PARTNER trial (J.E.A. et al., manuscript in preparation) showed significantly improved EFS and OS for patients in the research arm (gap schedule) compared with those in the control arm and even more so compared with those in the dropped arm (non-gap schedule).

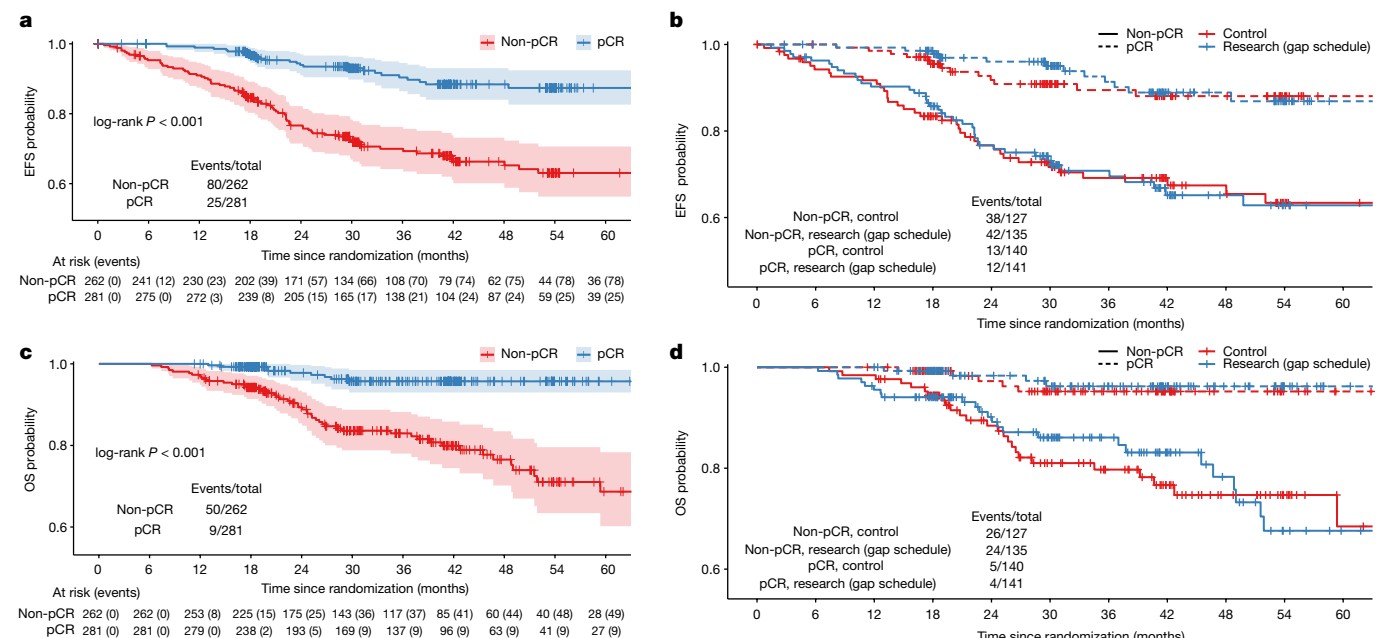

**Fig. 4 | Kaplan–Meier curves of survival by pathological response and treatment arm. a,b,** EFS by pathological response (**a**) and by pathological response and treatment arm (**b**). **c,d,** OS by pathological response (**c**) and by pathological response and treatment arm (**d**) in patients in the modified intention-to-treat group.

This confirms that olaparib was given at the optimal dose and schedule in combination with carboplatin–paclitaxel in the TNBC(gBRCAwt) cohort, and the lack of activity demonstrated in this group is not due to suboptimal dose or schedule. Pre-planned translational work has commenced on available samples in this cohort to establish whether there are smaller subgroups that can be identified with biomarkers of response to the gap-schedule olaparib.

There remains an important unanswered question on the utility of olaparib in early breast cancer, which the results from the PARTNER trial TNBC(gBRCAwt) cohort help to inform. It is not known whether patients with early TNBC(gBRCAwt) would benefit from adjuvant olaparib. To date, in this group there have been randomized controlled trials directly comparing adjuvant olaparib with the standard of care in patients with residual disease following neoadjuvant chemotherapy. However, as the PARTNER trial showed no hint of activity in the TNBC(gBRCAwt) cohort at this dose and schedule, and when this is compared with the major benefit of the same treatment in patients with gBRCAm breast cancer (J.E.A. et al., manuscript in preparation), it seems unlikely that there would be any effect for adjuvant olaparib treatment in the whole TNBC(gBRCAwt) group.

A strength of the PARTNER trial is the detailed, upfront characterization of patients' breast cancers. In addition, all patients were prospectively tested for gBRCA pathogenic variants and therefore in our first reporting here, we can confirm that all patients were gBRCAwt. In addition, all recruited patients had basal-like TNBC on the basis of immunohistochemistry assessments, and the patients in the non-basal group were excluded. Non-basal TNBCs, although triple negative, share few of the biological and genomic features of basal-like TNBC, and respond less well to most systemic treatments[20–22]. Therefore, their inclusion in randomized controlled trials of TNBC add non-informative data to the analyses that could affect results. Nevertheless, the designation of basal-like TNBC(gBRCAwt) by immunohistochemistry, although readily available in most centres, is not perfect and this group will undoubtedly include some tumours that are non-basal by molecular testing. Another strength is that the PARTNER trial is one of the few studies that can provide detailed safety data for the combination of chemotherapy and PARP inhibitor as a treatment for early-stage breast cancer. The combination was generally well tolerated and no significant differences

in treatment discontinuation due to toxicity were observed between groups. Of note however, more patients required a blood transfusion for treatment-induced anaemia with olaparib (51.4% versus 30.5%). High rates of symptomatic anaemia have also been described with the use of talazoparib monotherapy (39.3%) in the neoadjuvant setting[22]. Our data are therefore consistent with the known toxicity profile of PARP inhibitors, enhanced by the additional effect of platinum on the bone marrow. Despite this, the delivery of chemotherapy and olaparib was generally more than 90% and rates of pCR were high, comparable with the best from previously reported studies[10]. Notably, the GeparOLA trial reported overall low rates of grade 3–4 anaemia from olaparib (2.9%) and carboplatin (18.9%) in combination with paclitaxel[23]. This is probably explained by weekly dosing of carboplatin as well as the reduced dose of continuous olaparib.

Although the COVID-19 pandemic occurred during the trial, causing a pause in recruitment, the members of the teams at our recruiting centres and in the central Cambridge team ensured that enrolment into the TNBC(gBRCAwt) cohort was completed with minimum delay. Another strength was the inclusion of carboplatin in both the control

**Table 2 | Summary of AEs in the study period**

| | Research (gap schedule) (*n*=282) | Control (*n*=269) |
|---|---|---|
| Any AEs | 282 (100%) | 268 (99.6%) |
| AE grade ≥3 | 181 (64.2%) | 158 (58.7%) |
| Any SAE | 96 (34.0%) | 93 (34.6%) |
| SAE related to carboplatin | 60 (21.3%) | 49 (18.2%) |
| SAE related to paclitaxel | 63 (22.3%) | 47 (17.5%) |
| SAE related to olaparib | 45 (16.0%) | – |
| Missed doses due to toxicity | 114 (40.4%) | 76 (28.3%) |
| Modified doses due to toxicity | 109 (38.7%) | 48 (17.8%) |
| Treatment discontinued because of toxicity | 21 (7.4%) | 22 (8.2%) |
| Red cell transfusion required during chemotherapy | 145 (51.4%) | 82 (30.5%) |

and the research groups, which drove changes in the standard of care in the UK's National Health Service with recruiting centres adopting carboplatin-based chemotherapy early. In addition, now that the trial is completed it can inform adjuvant and neoadjuvant treatments for TNBC in 2024. Last, a major strength of the study is the strong and comprehensive translational research component (not reported here), with tumour tissue collection from all patients, and fresh tissue collection from patients treated in selected centres. In addition, patients from selected centres have had longitudinal circulating tumour DNA samples collected, and a cohort of patient-derived tumour xenografts have been developed in the Cambridge Centre. The translational research that is ongoing will be published later and should add to the knowledge base for TNBC(gBRCAwt) and accelerate the application of precision medicine and personalized treatment in this group of patients.

There were no major limitations in the study from a methodological or practical point of view. This study alongside the results from the gBRCAm cohort highlights the importance of the availability of rapid assessment of gBRCA pathogenic variants. The early knowledge of these biomarkers or their absence is now critical to ensure that patients with TNBC(gBRCAwt) and gBRCAm receive the most appropriate neoadjuvant regimens. Testing patients for germline pathogenic variants in *BRCA1*, *BRCA2* and also *PALB2* (ref. 24) in a clinically relevant time frame is essential to equip clinicians with the necessary information to apply a precision medicine approach.

The landscape of treatment for TNBC has changed considerably since planning for this study started in 2012. The CREATE-X trial demonstrated a significant benefit in patients with TNBC from 6 months of adjuvant capecitabine given after neoadjuvant chemotherapy that left residual disease[25]. The GEICAM–CIBOMA trial also addressed this question but showed no improvement with capecitabine in patients in the TNBC group as a whole, although it did report benefit in patients in the non-basal TNBC group[26]. Therefore, there is some evidence to change the standard of care to use adjuvant capecitabine for non-basal TNBC with residual disease after neoadjuvant chemotherapy.

The positive outcomes of KEYNOTE 522 (refs. 27,28) have resulted in the immune checkpoint inhibitor pembrolizumab becoming licensed for use as neoadjuvant and adjuvant therapy in TNBC by the US Food and Drug Administration[29], the European Medicines Evaluation Agency[30] and the National Institute for Health and Care Excellence[31]. KEYNOTE 522 has yet to report biomarker results defining the basal-like TNBC and gBRCAm cohorts, which would help to guide the standard of care for patients in these groups. However, pembrolizumab is now standard of care in early TNBC(gBRCAwt) in the neoadjuvant and adjuvant settings. Results from the NeoTRIP trial that tested neoadjuvant atezolizumab with chemotherapy[32] showed emerging signals of benefit from immunotherapy in patients with TNBC tumours with evidence of immune activation and remodelling in the tumour microenvironment[33]. Other published evidence confirms the prognostic importance of DNA-damage immune response signatures and stromal TILs[34]. Our analysis also confirmed that a high level of TILs (≥60%) correlated positively with increasing response rates to neoadjuvant treatments as has been shown by other groups[35], although olaparib did not seem to be more active in tumours with higher TIL counts.

## Conclusions

In conclusion, the PARTNER trial is a neoadjuvant study that has tested the addition of olaparib with a gap schedule to platinum-based chemotherapy in patients with basal-like TNBC who are known to be gBRCAwt. The trial did not show a benefit from the addition of olaparib in the dose and schedule used, either in rates of pCR or estimated EFS and OS. This is in marked contrast to the significant positive effect of the same dose and schedule in patients with gBRCAm breast cancer (J.E.A., manuscript in preparation). Further translational analysis will provide insights into the biology of TNBC(gBRCAwt) and may identify predictive biomarkers at baseline or after treatment to improve outcomes for patients with neoadjuvant olaparib.

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

¹Precision Breast Cancer Institute, Department of Oncology, University of Cambridge, Cambridge, UK. ²Cancer Research UK Cambridge Centre, University of Cambridge, Cambridge, UK. ³Cambridge Cancer Trials Centre, University of Cambridge, Cambridge, UK. ⁴Cambridge Cancer Trials Centre, Cambridge University Hospitals NHS Foundation Trust, Cambridge, UK. ⁵Department of Statistics, Athens University of Economics and Business, Athens, Greece. ⁶Royal Devon University Healthcare NHS Foundation Trust, Exeter, UK. ⁷Cambridge University Hospitals NHS Foundation Trust, Cambridge, UK. ⁸Cambridge Clinical Trials Unit, Cambridge University Hospitals NHS Foundation Trust, Cambridge, UK. ⁹The City Hospital, Nottingham University Hospitals NHS Trust, Nottingham, UK. ¹⁰Royal Bournemouth General Hospital, Bournemouth, UK. ¹¹Royal Wolverhampton NHS Trust, Wolverhampton, UK. ¹²Russells Hall Hospital, Dudley, UK. ¹³Ipswich Hospital, East Suffolk and North Essex NHS Foundation Trust, Ipswich, UK. ¹⁴University Hospital of Derby and Burton, Derby, UK. ¹⁵Bedford Hospital, Bedfordshire Hospitals NHS Foundation Trust, Bedford, UK. ¹⁶Pinderfields Hospital, Mid Yorkshire Teaching NHS Trust, Wakefield, UK. ¹⁷University Hospital Southampton NHS Foundation Trust, Southampton, UK. ¹⁸Basingstoke & North Hampshire Hospital, Basingstoke, UK. ¹⁹Royal Hampshire Hospital, Winchester, UK. ²⁰Velindre Cancer Centre, Cardiff, UK. ²¹University Hospitals Bristol and Weston NHS Foundation Trust, Bristol, UK. ²²Mount Vernon Cancer Centre, Northwood, UK. ²³Queens Hospital, Barking, Havering and Redbridge University Hospitals NHS Trust, Romford, UK. ²⁴Beatson West Of Scotland Cancer Centre, Glasgow, UK. ²⁵Swansea Bay University Health Board, Swansea, UK. ²⁶Hinchingbrooke Hospital, North West Anglia NHS Foundation Trust, Huntingdon, UK. ²⁷Macmillan Unit, West Suffolk Hospital NHS Foundation Trust, Bury Saint Edmunds, UK. ²⁸Worcestershire Acute Hospitals NHS Trust, Worcester, UK. ²⁹Alexandra Redditch Hospital, Redditch, UK. ³⁰Kidderminster Hospital, Kidderminster, Worcestershire, UK. ³¹Royal Free London NHS Foundation Trust, London, UK. ³²Oncology Department, Colchester General Hospital, East Suffolk & North Essex NHS Trust, Colchester, UK. ³³University Hospitals Dorset NHS Foundation Trust, Poole, UK. ³⁴University College London Hospitals NHS Foundation Trust, London, UK. ³⁵Department of Histopathology, Addenbrooke's Hospital, Cambridge University Hospitals NHS Foundation Trust, Cambridge, UK. ³⁶Oxford University Hospital NHS Foundation Trust, Oxford, UK. ³⁷Peterborough City Hospital, North West Anglia NHS Foundation Trust, Peterborough, UK. ³⁸The Christie NHS Foundation Trust and Division of Cancer Sciences, Manchester, UK. ³⁹Cancer Sciences Academic Unit, University of Southampton, Southampton, UK. ⁴⁰EMC2 Clinical Consultancy, Sale, Manchester, UK. ⁴¹Department of Medical Genetics, National Institute for Health Research, Cambridge Biomedical Research Centre, University of Cambridge, Cambridge, UK. ✉e-mail: ja344@cam.ac.uk

## Methods

### Patient and tumour characteristics

Patients aged between 16 and 70 years with histologically confirmed stage T1–4, N0–3 (tumour or axillary lymph node diameter ≥10 mm) invasive breast cancer, confirmed ER-negative and HER2-negative, and Eastern Cooperative Oncology Group performance status 0–1 were eligible. Other key inclusion criteria were patient fitness to receive the trial chemotherapy regimen and availability of slides and paraffin-embedded tissue blocks from the pretreatment biopsy. Patients were excluded if they had T0 tumour in the absence of axillary node ≥10 mm, apparent distant metastases, prior history of invasive breast cancer in the past 5 years or any previous chemotherapy or targeted agent used for the treatment of cancer in the past 5 years. The PARTNER trial protocol (NCT03150576) was approved by the North West – Haydock Research Ethics Committee (ref: 15/NW/0926) and the trial was carried out in accordance with the Declaration of Helsinki and the European Clinical Trials Directives 2001/20/EC. All patients provided an initial written informed consent that covered pathological review of the local slides and biopsy tissue at the Cambridge Centre, with additional biomarker review carried out centrally (EGFR, CK5, CK6 and AR) to confirm basal-like TNBC. If the biomarkers confirmed this, the patients proceeded to full consent for the main study at the local centre. Following trial entry, all patients were tested for pathogenic variants of germline *BRCA1* and *BRCA2* (gBRCAm). Those with gBRCAwt were included in this cohort, whereas those with gBRCAm were included in another cohort. The trial was sponsored by Cambridge University Hospitals NHS Foundation Trust and the University of Cambridge, and financed by a project grant from AstraZeneca, who also supplied olaparib. Cancer Research UK endorsed the trial and financed the sample collections for the translational studies that will be reported separately. The funders had no role in the study data collection, analysis, interpretation or writing of this report.

### Treatment

The trial was open label and was carried out in three stages. Stage 1 assessed the safety of combining olaparib with carboplatin and paclitaxel, and stage 2 compared two different schedules of olaparib and carboplatin–paclitaxel to 'pick-the-winner'. In stages 1 and 2, eligible patients were randomly assigned using a minimization method in a 1:1:1 ratio, with a web-based central randomization system. Patients in the control group received chemotherapy alone: carboplatin at area under the curve 5 (AUC5) intravenously on day 1 with paclitaxel 80 mg m$^{-2}$ intravenously on days 1, 8 and 15 every 3 weeks for four cycles. During stages 1 and 2, there were two randomized research arms in which intermittent dosing of olaparib at 150 mg twice a day by mouth (p.o) for 12 days was added to carboplatin–paclitaxel for each of four cycles. The schedule in the first research arm was olaparib at 150 mg twice a day on days −2 to day +10 of each of four cycles of carboplatin with paclitaxel and this was designated as the non-gap schedule. The schedule in the second research arm was olaparib at 150 mg twice a day by mouth from day +3 to day +14 and this was designated as the gap schedule. Patients were treated on a 3-weekly basis for four cycles in the control or research arms and then all patients had three cycles of standard local anthracycline-based chemotherapy without olaparib before surgery. The research arm (gap schedule) was subsequently selected and taken forwards to stage 3 and the non-gap schedule arm was the 'dropped arm'. This decision was based on the recommendation of the independent data safety monitoring committee (IDSMC), which based guidance on pre-specified criteria of safety, convenience and compliance, and efficacy.

In stage 3, patients were randomly assigned with a 1:1 ratio to either the control or research arm (gap schedule olaparib). Patients who had been randomized to the control arm and the gap schedule research arm in stages 1 and 2 were also included in this analysis. Stratification factors at randomization included tumour size (≤50 mm versus >50 mm), axillary lymph node involvement at diagnosis by biopsy and/or imaging (yes versus no) and TILs (<60% versus ≥60%). G-CSF was given as per local practice. A flow chart of the trial treatment is shown in Extended Data Fig. 1.

### Study procedures

Patients were clinically assessed before the beginning of every cycle until the end of treatment or disease progression. Breast surgery was carried out after 21 weeks of chemotherapy with or without olaparib and was followed by radiotherapy as per local standard protocols. After surgery, patients were followed 6-monthly for 2 years and then annually for up to 10 years. Local histopathology reports from primary surgery were centrally reviewed independently by two readers (clinicians and pathologists) blinded to the treatment allocation, for each report (E.P., H.M.E., L.M.D. and A.F.), and if there were any differences in any of the response criteria (Extended Data Fig. 7), consensus for each patient was reached after discussion. AEs were reported for each cycle of protocol treatment using NCI CTCAE version 4.03. Participation in the quality of life (QOL) sub-study was optional and this metric was assessed using two measures. The first was the EQ-5D-5L measure, which is a self-report survey that measures QOL across 5 domains: mobility, self-care, usual activities, pain/discomfort and anxiety/depression. The second was the FACT-B, which is a 37-item instrument designed to measure five domains of HRQOL in patients with breast cancer. Both questionnaires were completed by patients before randomization, following completion of four cycles, completion of seven cycles, surgery and radiotherapy, and annually for 2 years from completion of surgery.

### Statistical analysis

In this three-stage phase II–III trial, stage 1 assessed the safety of the addition of olaparib to 3-weekly carboplatin with weekly paclitaxel chemotherapy. Stage 2 selected the 'winner' from two research arms, and stage 3 assessed pCR at surgery after neoadjuvant treatment in all patients. The primary end point was comparison of pCR rates between the research and control groups. Details of the statistical analysis plan for each stage are provided in the Supplementary Information. Secondary survival end points were all calculated from the date of randomization to the date of first event and included: EFS (local or distant recurrence, diagnosis of a second cancer or death from any cause); relapse-free survival (local or distant recurrence or death from any cause, excluding patients who relapsed before surgery); breast cancer-specific survival (death from breast cancer or death after breast cancer relapse); DDFS (distant recurrence or death from any cause); local recurrence-free survival (local recurrence or death from any cause); OS (death from any cause); time to second cancer (diagnosis of a second cancer). Other secondary end points were: residual cancer burden; pCR in breast alone; radiological response; safety and quality of life.

In this TNBC(gBRCAwt) cohort, a total of 454 patients were needed to test with 90% power and 5% significance level the null hypothesis of no difference in pCR rate between the two groups, versus the alternative of 50% in the control group and 65% in the research group. Allowing for a non-compliance of 5%, it was planned to recruit a total of 478 patients TNBC(gBRCAwt) between the control and the selected research group.

The main analysis was conducted on the basis of the modified intention-to-treat principle, which included all randomized, eligible patients excluding only those who did not start treatment. The safety analyses included patients who had at least one dose of trial treatment. The differences between binary outcomes were compared using the chi-squared test and the CI of the differences was based on the score method[36]. Kaplan–Meier plots were generated for time-to-event outcomes and groups were compared using the log-rank test. HRs were estimated using the Cox regression model. The subscales of the EQ-5D-5L and FACT-B questionnaires were derived according to standard-scoring

manuals. Analyses of changes from the baseline over time and differences between the two groups for subscales were carried out with repeated measures analysis of covariance, adjusting for the baseline level, time, treatment and interaction of time and treatment. Although it was assumed that the data were 'missing at random', a sensitivity analysis for data 'missing not at random' was carried out for the primary end point. All statistical analyses were carried out in R (v4.1.0) and all $P$ values are based on two-tailed tests.

## Reporting summary

Further information on research design is available in the Nature Portfolio Reporting Summary linked to this article.

## Data availability

Data collected in the PARTNER study will be made available to researchers whose full proposal for their use of the data has been approved by the PARTNER trial management group and whose research includes a clear and comprehensive research plan with statistical considerations adequately completed. The data required for the approved, specified purposes and the trial protocol will be provided after completion of a data sharing agreement. Data sharing agreements will be set up by the trial steering and management groups and will include clear instructions on publication, reporting and usage policy. A minimum dataset of anonymized data will be made available after full publication of the trial and related work. Requests for data should be addressed to ja344@cam.ac.uk.

36. Miettinen, O. & Nurminen, M. Comparative analysis of two rates. *Stat. Med.* **4**, 213–226 (1985).

**Acknowledgements** We thank the patients, and the families and friends who supported them, for participating in this trial; our ethics committee, our independent data and safety monitoring committee and the trial management group for their advisory roles; the PARTNER trial consortium members, past and present (Supplementary Information); and I. Cizaite for preparing and proofreading the manuscript. This trial was sponsored by Cambridge University Hospitals NHS Foundation Trust and the University of Cambridge, and financed by a project grant from AstraZeneca, who also supplied olaparib. Cancer Research UK provided peer review and endorsement for the study and financed the sample collections for the translational studies, which will be reported separately. We also acknowledge the National Institute for Health and Care Research Cambridge Biomedical Research Centre and the Cancer Research UK Cambridge Centre for their financial support for staff and infrastructure costs. The funders had no role in data collection or analysis. Once the trial group had interpreted the data, the results were then shared with the AstraZeneca scientists. In addition, we thank the Cancer Molecular Diagnostics Laboratory and The Precision Breast Cancer Institute Team for their support for sample collection; Cambridge Tissue Bank (NIHR203312) for sample assessment and diagnostics; Cambridge Clinical Trials Centre – Cancer Theme for their core staff support; the clinical trials support staff at all participating sites; and Addenbrookes Charitable Trust for financing the post of the chief investigator (2015–2018). We acknowledge Cancer Research UK (CRUKE/14/048) and AstraZeneca (1994-A093777).

**Author contributions** J.E.A.: chief investigator, overall oversight, design, protocol development, management, data interpretation and manuscript writing; site principal investigator for co-ordinating central site. K.P.: clinical fellow contributing to manuscript writing, data interpretation, trial safety management and protocol development. L.G.: senior trials co-ordinator dealing with day-to-day management of the trial and all the trial team. A.D.: main statistician dealing with data analysis. N.D.: senior statistician overseeing data analysis. C.H.: senior data manager overseeing data collection team. L.M.D.: clinical fellow, contributing to trial safety management and pathology report assessment. R.L.: clinical fellow, contributing to trial safety management and protocol development. A.F.: clinical fellow, contributing to trial safety management and pathology report assessment. A.N.R.: clinical trials practitioner and contributed to case report development. J.R.W.: senior clinical trials practitioner and contributed to case report development and sample management. A. Chabbra: senior pharmacist contributed to trial development. W.Q.: original lead statistician. A.-L.V.: original senior trials co-ordinator. R.M.H.: lead programmer for development of case report form and data collection proformas. Site principal investigators (recruitment and consent of patients): S.C., T.H., D.T., R.V., M.P., S.A., D.G., S.R., A.B., J.P.B., S.S., E.S., L.C.S., M.D., C.A.P., M.M., M.J.C., M.B.M., A. Chakrabarti, R.R.R. and N.C.L. Site principal investigators with trial management group responsibilities: J.C.N., K.M., A.C.A. (co-chair of trial management group), E.R.C. (co-chair of trial management group). P.C.S.: trial pathologist. E.M.: trial development and support. M.T.: medical genetics lead. E.P.: senior trial pathologist. H.M.E.: design, management, data interpretation and manuscript writing.

**Competing interests** The authors declare competing financial interests in the form of research grants and honoraria for lectures. The funders of the research grants and honoraria had no role in the study design, data collection, analysis, interpretation or writing of the manuscript. J.E.A. reports honoraria, conference attendance travel support and a grant from AstraZeneca; and honoraria from Esai and Pfizer for lectures. M.M. reports shares in AstraZeneca. M.B.M. reports advisory board membership of Roche, Pfizer, MSD, Daiichi Sankyo, Gilead, AstraZeneca, Novartis, the Menarini Group, Genomic Health (Precision Medicine) and Seagen; speaker honoraria from Roche, BMS, Seagen, Pfizer, Daiichi Sankyo, AstraZeneca, Lilly, MSD, Genomic Health (Precision Medicine), Eisai and Novartis; and meeting expenses from Roche, Eli Lilly, Novartis and MSD. R.R.R. reports honoraria from Daiichi Sankyo, AstraZeneca, Novartis and Pfizer; membership of advisory boards for Daiichi Sankyo, Eli Lilly, Pfizer and AstraZeneca; and travel and conference attendance from BMS, Pfizer and Roche. N.C.L. reports that their partner is employed by AstraZeneca. N.C.L. reports shares in AstraZeneca. A.C.A. reports research funding paid to their institution from AstraZeneca; conference fees and travel expenses from Roche and Novartis; conference fees from MSD; membership of Roche and AstraZeneca advisory boards; and a grant for an educational project from Gilead. E.R.C. reports honoraria from AstraZeneca, Eli Lilly, Novartis, Pfizer and Roche; membership of advisory boards for AstraZeneca, Eli Lilly, Pfizer, Menarini Stemline UK and Novartis; consultancy for Pfizer; conference fees, travel and accommodation from Roche and Novartis; an educational grant from Daiichi Sankyo; and research funding and support from SECA and AstraZeneca. E.P. reports honoraria from Roche, Novartis and AstraZeneca. The remaining authors declare no competing interests.

**Additional information**
**Correspondence and requests for materials** should be addressed to Jean E. Abraham.

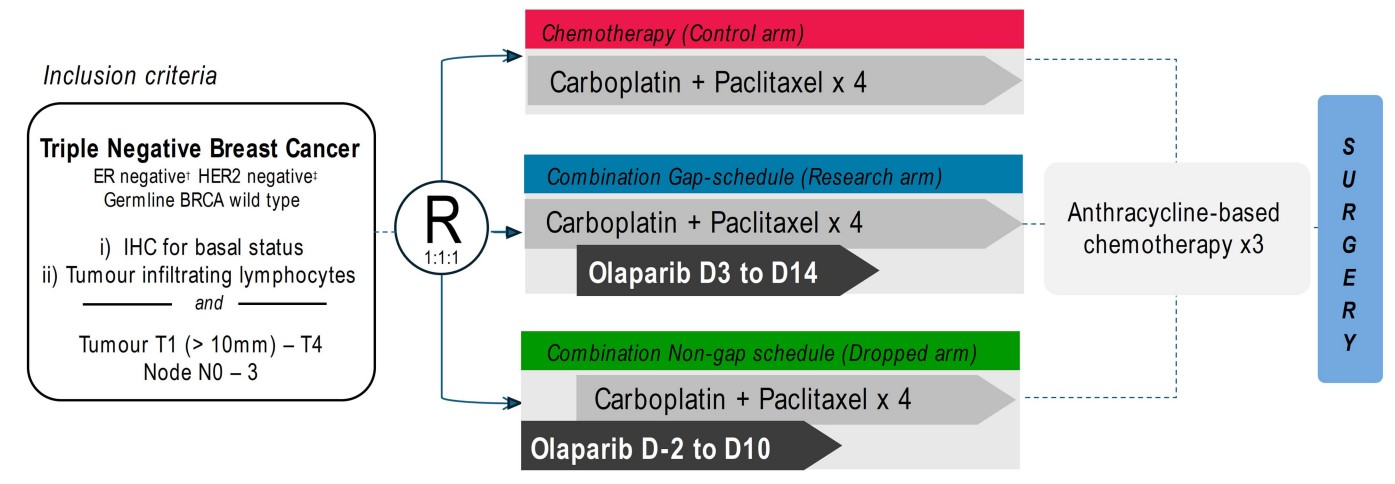

Inclusion criteria

**Triple Negative Breast Cancer**
ER negative† HER2 negative‡
Germline BRCA wild type

i) IHC for basal status
ii) Tumour infiltrating lymphocytes
———— *and* ————

Tumour T1 (> 10mm) – T4
Node N0 – 3

**R** 1:1:1

*Chemotherapy (Control arm)*
Carboplatin + Paclitaxel x 4

*Combination Gap-schedule (Research arm)*
Carboplatin + Paclitaxel x 4
**Olaparib D3 to D14**

*Combination Non-gap schedule (Dropped arm)*
Carboplatin + Paclitaxel x 4
**Olaparib D-2 to D10**

Anthracycline-based chemotherapy x3

**S U R G E R Y**

†ER negative defined as nuclear staining <10% or AS 0-3. ‡HER2 negative defined as IHC 0,1+ or IHC 2+ ISH not amplified. Stratification factors: Histopathological involvement of axillary nodes, Tumour size <=50mm, >50mm, TILs <=60, >60%
Non-gap scheduled arm was dropped after completion of Stage 2. Randomization 1:1 after arm selection

Paclitaxel 80mg/m2 Day 1,8,15 every 3 weeks
Carboplatin AUC5 Day 1 every 3 weeks
Olaparib 150mg twice daily Day -2 to Day 10 or Day 3 to Day 14 every three weeks

**Extended Data Fig. 1 | Trial Flow Chart: PARTNER Trial for the TNBC Cohort.** Trial Flow Chart: PARTNER Trial for the TNBC Cohort.

| Subgroup | Research (gap scheduled) | Control | Difference in pCR (95% CI) | |
|---|---|---|---|---|
| | no. of patients with response/no. of patients (%) | | percentage point | |
| **Overall** | **141/276 (51.1)** | **140/267 (52.4)** | | **-1.3 (-9.7 to 7.0)** |
| **Age** | | | | |
| ≤ 40 | 101/198 (51.0) | 98/186 (52.7) | | -1.7 (-11.6 to 8.3) |
| (40, 65] | 32/59 (54.2) | 29/59 (49.2) | | 5.1 (-12.9 to 22.7) |
| >65 | 8/19 (42.1) | 13/22 (59.1) | | -17.0 (-44.8 to 13.8) |
| **Tumour size** | | | | |
| ≤ 50mm | 134/261 (51.3) | 137/256 (53.5) | | -2.2 (-10.7 to 6.4) |
| > 50mm | 7/15 (46.7) | 3/11 (27.3) | | 19.4 (-19.1 to 51.6) |
| **Axillary Lymph Node Involvement at diagnosis by Biopsy and/or Imaging** | | | | |
| No | 106/187 (56.7) | 100/187 (53.5) | | 3.2 (-6.9 to 13.2) |
| Yes | 35/89 (39.3) | 40/80 (50.0) | | -10.7 (-25.3 to 4.4) |
| **Tumour Infiltrating Lymphocytes TILs Score** | | | | |
| < 60% | 97/211 (46.0) | 104/209 (49.8) | | -3.8 (-13.3 to 5.8) |
| ≥ 60% | 44/65 (67.7) | 36/58 (62.1) | | 5.6 (-11.2 to 22.4) |

-40  -20  0  20  40

Control Better   Research Better

**Extended Data Fig. 2 | Forest plot of the primary end-point in subgroups.** The data are the numbers and proportions of patients with pathological complete response (pCR). Data were analysed from a total of 276 in research (gap schedule) and 267 control arms. Estimates are the difference in the pCR and 95% CI based on the score method between research arm and control arm.

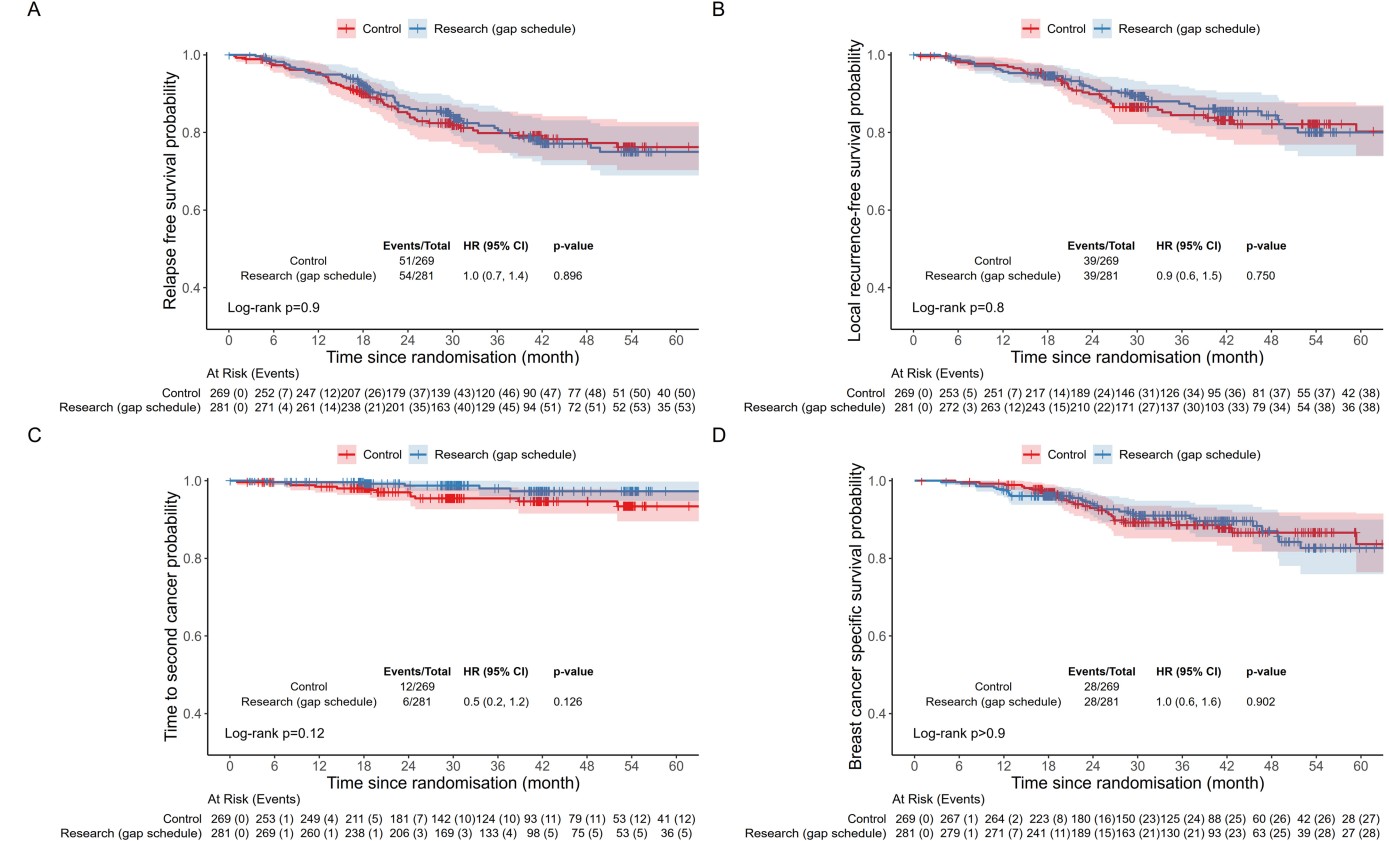

**Extended Data Fig. 3 | Additional Kaplan-Meier curves of survival endpoints (part 1).** Relapse free survival (A), local recurrence free survival (B), time-to-second cancer (C) and breast cancer specific survival (D) by treatment arm (control and research arm).

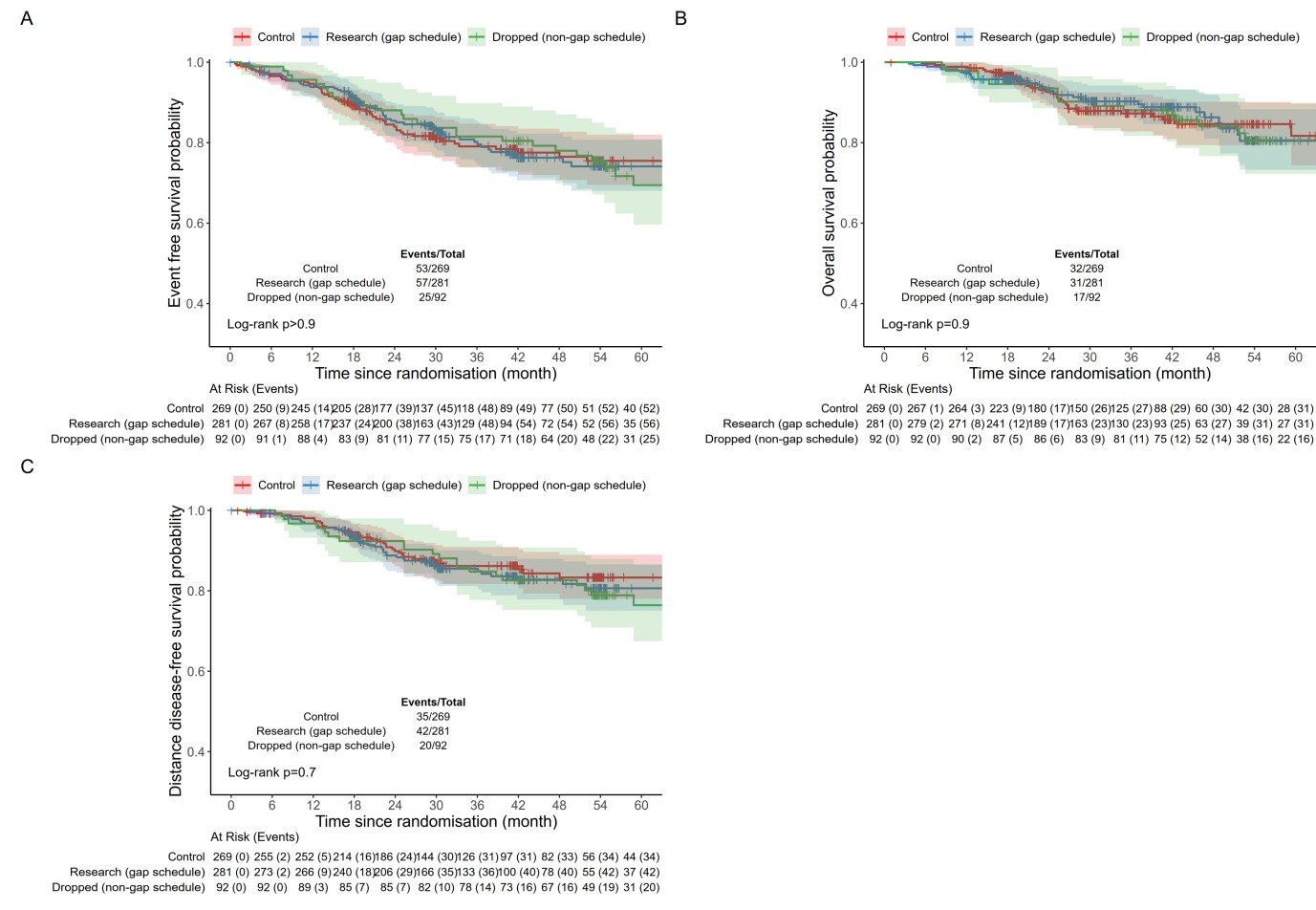

**Extended Data Fig. 4 | Additional Kaplan-Meier curves of survival endpoints (part 2).** Relapse free survival (A), overall survival (B), and distant disease-free survival (C) by treatment arm (control, research arm, and dropped arm) in all patients.

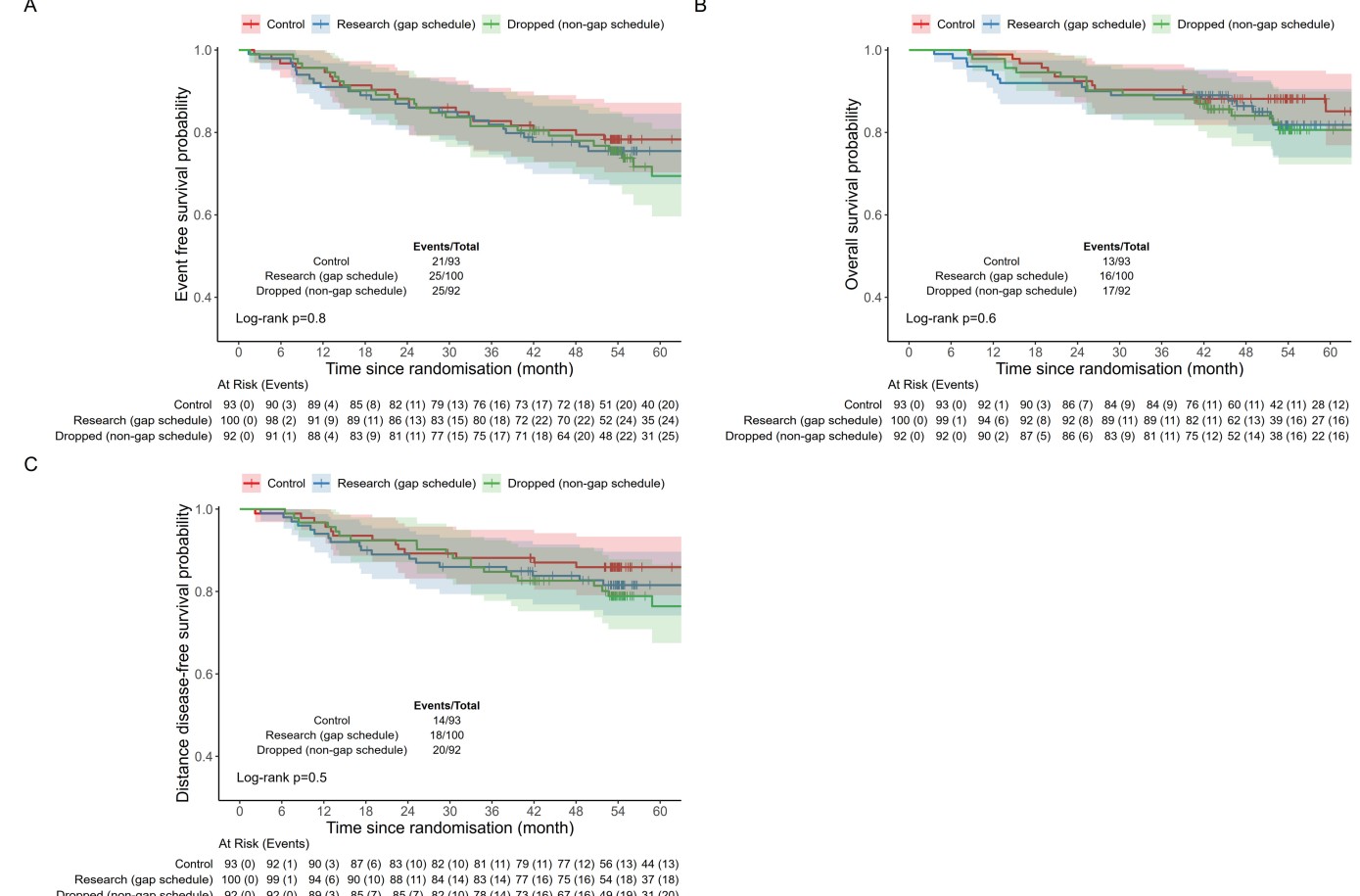

**Extended Data Fig. 5 | Additional Kaplan-Meier curves of survival endpoints (part 3).** Relapse free survival (A), overall survival (B), and distant disease-free survival (C) by treatment arm [control, research arm (gap-scheduled), and dropped arm (non-gap-scheduled)) in patients who were assessed at stage II.

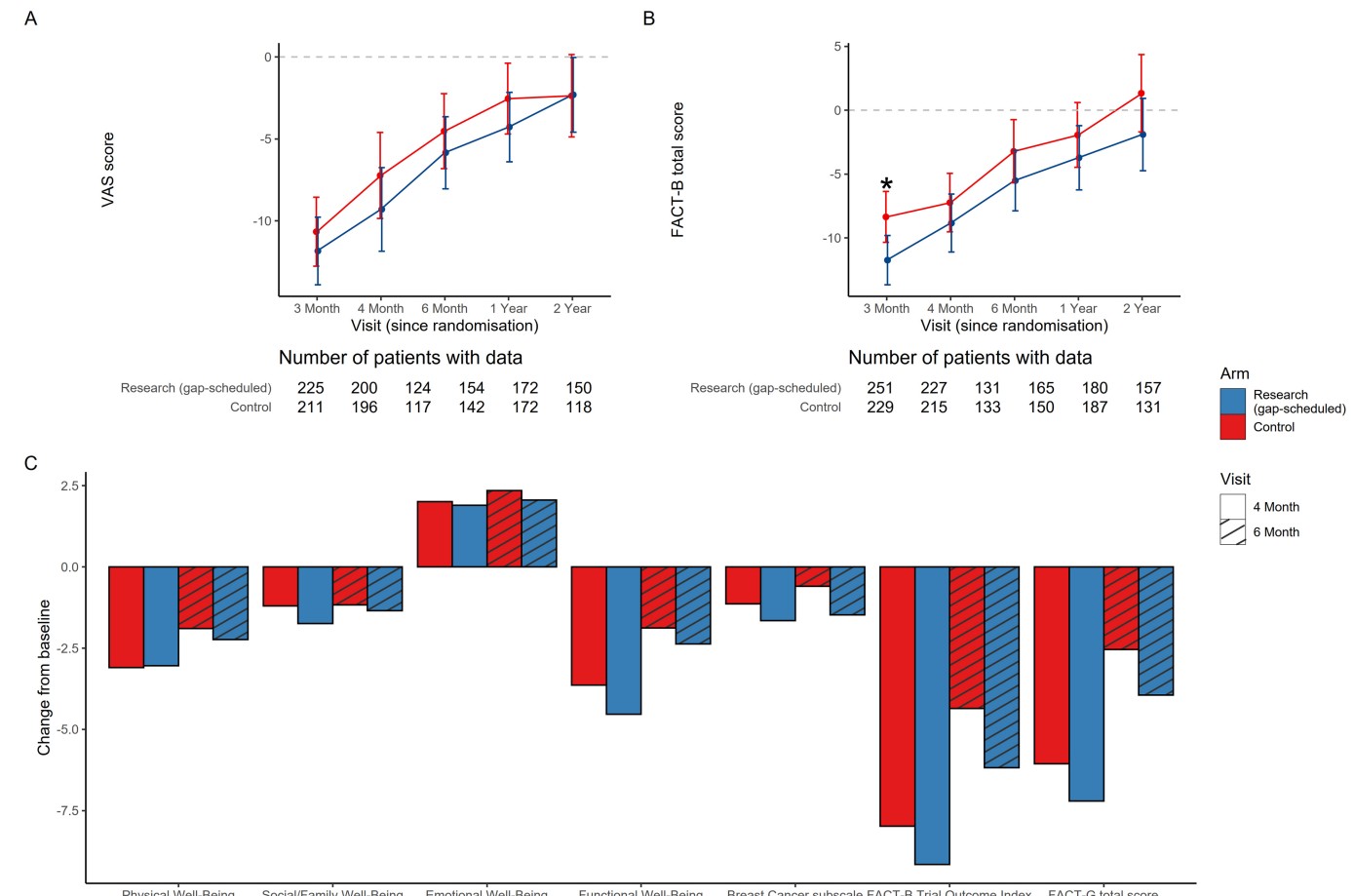

**Extended Data Fig. 6 | EQ-5D-5L visual analogue scale (VAS) scores and FACT-B total scores.** Least-squares mean change from baseline of EQ-5D-5L visual analogue scale (VAS) score (A), FACT-B total score (B) and FACT-B subscales with trial outcome index and FACT-B total score at 4 months and 6 months. Asterisk indicates a statistical difference between the treatment arms of p < 0.05 based on two-tailed t-test with Tukey's adjustment for multiple comparisons. Error bars, 95% CI of the estimated change score.

## Form page 1 of 3

**Patient Initials:** ☐☐  **Site Code:** N 0 0 ☐☐☐☐☐
**Date of Birth:** ☐☐/☐☐/☐☐☐☐  **Trial No.:** 1 ☐☐☐

**Central Pathology Review**  *Form page 1 of 3*  **23**

| | |
|---|---|
| Pathology Hospital | ☐ |
| Pathology Report Numbers (if available) | ☐ |
| Person completing form | ☐  1 = Reviewer 1 / 2 = Reviewer 2 / 3 = Reviewer 1 and 2 |
| Date of assessment | ☐☐/☐☐/☐☐☐☐ |

### Breast 1 pathCR Assessment

| | |
|---|---|
| Breast Site | ☐  1 = Left / 2 = Right |
| Response in breast | ☐  1 = pCR / 2 = non-pCR / 3 = Not assessable |
| If pCR, was tumour bed identified? | ☐  0 = No / 1 = Yes / 2 = No comment |
| Residual DCIS? | ☐  0 = No / 1 = Yes / 2 = Unknown |

**Axillary Lymph Node Assessment**

| | |
|---|---|
| Were axillary nodes examined? | ☐  0 = No / 1 = Yes |

*If yes, please answer all applicable questions in this section*

| | |
|---|---|
| Does the report state the number of nodes examined? | ☐  0 = No / 1 = Yes |

*If yes, please answer the following two questions*

| | |
|---|---|
| Number of nodes examined | ☐ |
| Number of nodes involved | ☐ |
| Response in axilla | ☐  1 = pCR / 2 = non-pCR / 3 = Not assessable |
| Evidence of fibrosis in nodes? | ☐  0 = No / 1 = Yes / 2 = No comment / 3 = N/A |

**Name:** ☐  **Signature:** ☐  **Date:** ☐☐/☐☐/☐☐☐☐

Authorised personnel should complete and send the ORIGINAL completed, signed and dated form (keep a photocopy at site) to:
PARTNER Trial Coordinator, Cambridge Clinical Trials Unit – Cancer Theme, Box 279 (S4), Addenbrooke's Hospital, Hills Road, Cambridge, CB2 0QQ

**For CCTU use only** ☐☐☐☐☐☐☐  Generated by CCTU-Cancer Theme's CRFGen  **Version 6.0 – 06 Jan 2022**

---

## Form page 2 of 3

**Patient Initials:** ☐☐  **Site Code:** N 0 0 ☐☐☐☐☐
**Date of Birth:** ☐☐/☐☐/☐☐☐☐  **Trial No.:** 1 ☐☐☐

**Central Pathology Review**  *Form page 2 of 3*  **23**

**Overall Assessment**

| | |
|---|---|
| Response in breast and axilla | ☐  1 = pCR / 2 = non-pCR / 3 = Not assessable |
| Does further pathological information need requesting? | ☐  0 = No / 1 = Yes |
| Comments | ☐ |

**Name:** ☐  **Signature:** ☐  **Date:** ☐☐/☐☐/☐☐☐☐

Authorised personnel should complete and send the ORIGINAL completed, signed and dated form (keep a photocopy at site) to:
PARTNER Trial Coordinator, Cambridge Clinical Trials Unit – Cancer Theme, Box 279 (S4), Addenbrooke's Hospital, Hills Road, Cambridge, CB2 0QQ

**For CCTU use only** ☐☐☐☐☐☐☐  Generated by CCTU-Cancer Theme's CRFGen  **Version 6.0 – 06 Jan 2022**

---

## Form page 3 of 3

**Patient Initials:** ☐☐  **Site Code:** N 0 0 ☐☐☐☐☐
**Date of Birth:** ☐☐/☐☐/☐☐☐☐  **Trial No.:** 1 ☐☐☐

**Central Pathology Review**  *Form page 3 of 3*  **23**

### Breast 2 pathCR Assessment (only to be completed for bilateral disease)

| | |
|---|---|
| Breast Site | ☐  1 = Left / 2 = Right |
| Response in breast | ☐  1 = pCR / 2 = non-pCR / 3 = Not assessable |
| If pCR, was tumour bed identified? | ☐  0 = No / 1 = Yes / 2 = No comment |
| Residual DCIS? | ☐  0 = No / 1 = Yes / 2 = Unknown |

**Axillary Lymph Node Assessment**

| | |
|---|---|
| Were axillary nodes examined? | ☐  0 = No / 1 = Yes |

*If yes, please answer all questions applicable in this section*

| | |
|---|---|
| Does the report state the number of nodes examined? | ☐  0 = No / 1 = Yes |

*If yes, please answer the following two questions*

| | |
|---|---|
| Number of nodes examined | ☐ |
| Number of nodes involved | ☐ |
| Response in axilla | ☐  1 = pCR / 2 = non-pCR / 3 = Not assessable |
| Evidence of fibrosis in nodes? | ☐  0 = No / 1 = Yes / 2 = No comment / 3 = N/A |

**Overall Assessment**

| | |
|---|---|
| Response in breast and axilla | ☐  1 = pCR / 2 = non-pCR / 3 = Not assessable |
| Does further pathological information need requesting? | ☐  0 = No / 1 = Yes |
| Comments | ☐ |

**Name:** ☐  **Signature:** ☐  **Date:** ☐☐/☐☐/☐☐☐☐

Authorised personnel should complete and send the ORIGINAL completed, signed and dated form (keep a photocopy at site) to:
PARTNER Trial Coordinator, Cambridge Clinical Trials Unit – Cancer Theme, Box 279 (S4), Addenbrooke's Hospital, Hills Road, Cambridge, CB2 0QQ

**For CCTU use only** ☐☐☐☐☐☐☐  Generated by CCTU-Cancer Theme's CRFGen  **Version 6.0 – 06 Jan 2022**

---

**Extended Data Fig. 7 | Central pathology review Case Report Form.** Central pathology review Case Report Form.

**Extended Data Table 1 | Summary of treatment received during the first regimen by treatment arm**

| Variable | Research (gap schedule) (N=281) | Control (N=269) | Total (N=550) |
|---|---|---|---|
| **Carboplatin** | | | |
| **Total Doses given, mg** | | | |
| Mean (SD) | 2430 (453) | 2370 (478) | 2400 (466) |
| Median [Min, Max] | 2500 [0, 3160] | 2400 [400, 3160] | 2450 [0, 3160] |
| **Total Duration, week** | | | |
| Mean (SD) | 12.5 (3.26) | 11.9 (1.70) | 12.2 (2.64) |
| Median [Min, Max] | 12.0 [6.00, 64.1] | 12.0 [3.00, 15.4] | 12.0 [3.00, 64.1] |
| Missing | 1 (0.4%) | | 1 (0.2%) |
| **Dose Intensity, mg/week** | | | |
| Mean (SD) | 196 (37.6) | 200 (31.5) | 198 (34.8) |
| Median [Min, Max] | 198 [0, 263] | 201 [127, 263] | 200 [0, 263] |
| **Planned dose received >80%*** | | | |
| Yes | 270 (96.4%) | 253 (94.1%) | 523 (95.3%) |
| No | 10 (3.6%) | 16 (5.9%) | 26 (4.7%) |
| Missing | 1 | | 1 |
| **Olaparib** | | | |
| **Total Doses given, mg** | | | |
| Mean (SD) | 13000 (3080) | | 13000 (3080) |
| Median [Min, Max] | 14400 [0, 15600] | | 14400 [0, 15600] |
| **Total Duration, week** | | | |
| Mean (SD) | 12.2 (3.19) | | 12.2 (3.19) |
| Median [Min, Max] | 12.0 [3.00, 58.1] | | 12.0 [3.00, 58.1] |
| Missing | 2 (0.7%) | | 2 (49.3%) |
| **Dose Intensity, mg/week** | | | |
| Mean (SD) | 1080 (223) | | 1080 (223) |
| Median [Min, Max] | 1170 [0, 1300] | | 1170 [0, 1300] |
| **Planned dose received >80%** | | | |
| Yes | 247 (88.2%) | | 247 (88.2%) |
| No | 33 (11.8%) | | 33 (11.8%) |
| Missing | 1 | | 1 |
| **Paclitaxel** | | | |
| **Total Doses given, mg** | | | |
| Mean (SD) | 1540 (285) | 1550 (358) | 1550 (322) |
| Median [Min, Max] | 1580 [132, 1980] | 1580 [0, 2160] | 1580 [0, 2160] |
| **Total Duration, week** | | | |
| Mean (SD) | 12.0 (3.40) | 11.8 (3.64) | 11.9 (3.52) |
| Median [Min, Max] | 12.0 [1.00, 61.1] | 12.0 [1.00, 60.1] | 12.0 [1.00, 61.1] |
| Missing | | 1 (0.4%) | 1 (0.2%) |
| **Dose Intensity, mg/m2/week** | | | |
| Mean (SD) | 130 (20.9) | 133 (23.8) | 131 (22.4) |
| Median [Min, Max] | 132 [28.3, 180] | 133 [0, 180] | 132 [0, 180] |
| **Planned dose received >80%** | | | |
| Yes | 281 (100%) | 265 (98.9%) | 546 (99.5%) |
| No | 0 | 3 (1.1%) | 3 (0.5%) |
| Missing | | 1 | 1 |

*Planned dose for carboplatin was based on derived dose using the Calvert formula.

**Extended Data Table 2 | Logistic regression analysis of primary endpoint imputing missing data with delta-adjusted pattern mixture model**

| Delta | OR (95% CI) | p-value (two-sided) |
|---|---|---|
| log(0.4) | 0.9 (0.7, 1.3) | 0.687 |
| log(0.6) | 0.9 (0.7, 1.3) | 0.701 |
| log(0.8) | 0.9 (0.7, 1.3) | 0.698 |
| log(1) | 0.9 (0.7, 1.3) | 0.735 |
| log(1.2) | 1.0 (0.7, 1.3) | 0.773 |
| log(1.4) | 0.9 (0.7, 1.3) | 0.759 |
| log(1.6) | 1.0 (0.7, 1.3) | 0.772 |
| log(1.8) | 1.0 (0.7, 1.3) | 0.772 |
| log(2) | 0.9 (0.7, 1.3) | 0.745 |

The imputation was based on delta-adjusted pattern mixture model adjusting for stratification factors with a range of delta values. The pattern mixture model of a binary endpoint can be expressed in the following form:

$\text{logit}[\Pr(Y=1|R_y, X)] = \gamma_0 + \gamma_1 X + \delta(1 - R_y)$

Where $\delta$ represents the difference in the log-odds of $Y=1$ between non-missing and missing, $X$ are observed covariates, which is stratification factors here, and $R_y$ is a missing indicator of $Y$. While $\delta = 0$ implies data missing at random, $\delta > 0$ implies better outcome in missing patients and $\delta < 0$ otherwise. All the missing pCR was imputed with the same $\delta$. For example, log(0.5), log(1) and log(1.5) indicates patients with missing pCR would have an odds of 0.5, 1 and 1.5 times achieving a pCR compared to non-missing patients.

**Extended Data Table 3 | Summary of events observed during the trial follow-up**

| Event | Research (gap schedule) | Control |
|---|---|---|
| Relapse | 19 | 14 |
| Progression on trial treatment | 6 | 3 |
| Progression after completion of trial treatment | 27 | 23 |
| First second primary | 6 | 12 |
| Death | 31 | 32 |

Note: each patient may have multiple events during the follow-up.

Relapse: patient completely responded to trial treatment but relapsed after trial treatment completed.

Progression on trial treatment: patient's disease has progressed during trial treatment.

Progression after completion of trial treatment: patient's disease did not completely respond to trial treatment and progressed after trial treatment finished.

First second primary: the first occurrence of a new primary cancer diagnosed after randomisation.

**Extended Data Table 4 | Summary of time-to-event outcomes**

| | Research (gap schedule) (N=281) | Control (N=269) |
|---|---|---|
| **Event free survival** | | |
| Events | 57 | 53 |
| Median (95% CI) | NE | NE |
| 36 months survival rate (95% CI) | 80.2 (75.2, 85.5) | 79.1 (73.9, 84.7) |
| Cox model HR (95% CI) | 1.0 (0.7, 1.4); 0.947 | |
| **Overall survival** | | |
| Events | 31 | 32 |
| Median (95% CI) | NE | 74.7 (74.7, NE) |
| 36 months survival rate (95% CI) | 90.3 (86.5, 94.2) | 87.2 (82.8, 91.9) |
| Cox model HR (95% CI) | 0.9 (0.6, 1.5); 0.797 | |
| **Distance disease-free survival** | | |
| Events | 42 | 35 |
| Median (95% CI) | NE | NE |
| 36 months survival rate (95% CI) | 85.5 (81.2, 90.1) | 86.2 (81.7, 90.9) |
| Cox model HR (95% CI) | 1.1 (0.7, 1.7); 0.632 | |
| **Relapse free survival** | | |
| Events | 54 | 51 |
| Median (95% CI) | NE | NE |
| 36 months survival rate (95% CI) | 81.1 (76.2, 86.4) | 79.9 (74.7, 85.4) |
| Cox model HR (95% CI) | 1.0 (0.7, 1.4); 0.896 | |
| **Local recurrence-free survival** | | |
| Events | 39 | 39 |
| Median (95% CI) | NE | NE |
| 36 months survival rate (95% CI) | 87.4 (83.2, 91.8) | 84.5 (79.7, 89.5) |
| Cox model HR (95% CI) | 0.9 (0.6, 1.5); 0.750 | |
| **Time to second cancer** | | |
| Events | 6 | 12 |
| Median (95% CI) | NE | NE |
| 36 months survival rate (95% CI) | 98.0 (96.0, 100.0) | 95.4 (92.7, 98.3) |
| Cox model HR (95% CI) | 0.5 (0.2, 1.2); 0.126 | |
| **Breast cancer specific survival** | | |
| Events | 28 | 28 |
| Median (95% CI) | NE | 74.7 (74.7, NE) |
| 36 months survival rate (95% CI) | 91.0 (87.4, 94.8) | 88.6 (84.3, 93.1) |
| Cox model HR (95% CI) | 1.0 (0.6, 1.6); 0.902 | |

NE: not estimated.

Extended Data Table 5 | Summary of AE grade ≥ 3 experienced per patient by term and grade in at least 1% of non-BRCA patients

| CTCAE v4.03 | Grade | Research (gap schedule) (N=282) | Control (N=269) | P* |
|---|---|---|---|---|
| Neutrophil count decreased no associated fever | 3 | 95 (33.7%) | 50 (18.6%) | 0.002 |
|  | 4 | 29 (10.3%) | 23 (8.6%) | 0.880 |
| Anaemia | 3 | 68 (24.1%) | 43 (16.0%) | 0.127 |
|  | 4 | 1 (0.4%) | 0 (0%) | 1.000 |
| Fatigue | 3 | 27 (9.6%) | 28 (10.4%) | 0.974 |
| Febrile neutropenia | 3 | 23 (8.2%) | 20 (7.4%) | 0.974 |
|  | 4 | 3 (1.1%) | 5 (1.9%) | 0.818 |
| Platelet count decreased | 3 | 29 (10.3%) | 11 (4.1%) | 0.084 |
|  | 4 | 3 (1.1%) | 3 (1.1%) | 1.000 |
| White blood cell decreased | 3 | 19 (6.7%) | 13 (4.8%) | 0.818 |
|  | 4 | 2 (0.7%) | 7 (2.6%) | 0.471 |
| Hypertension | 3 | 12 (4.3%) | 11 (4.1%) | 1.000 |
| Diarrhoea | 3 | 12 (4.3%) | 6 (2.2%) | 0.818 |
|  | 4 | 1 (0.4%) | 0 (0%) | 1.000 |
| Alanine aminotransferase increased | 3 | 10 (3.5%) | 6 (2.2%) | 0.818 |
|  | 4 | 1 (0.4%) | 0 (0%) | 1.000 |
| Thromboembolic event | 3 | 11 (3.9%) | 5 (1.9%) | 0.818 |
| Oral mucositis | 3 | 7 (2.5%) | 8 (3.0%) | 0.974 |
| Dyspnea | 3 | 7 (2.5%) | 5 (1.9%) | 0.974 |
| Nausea | 3 | 8 (2.8%) | 4 (1.5%) | 0.818 |
| Sensory neuropathy | 3 | 10 (3.5%) | 2 (0.7%) | 0.204 |
| Syncope | 3 | 3 (1.1%) | 6 (2.2%) | 0.818 |
| Vomiting | 3 | 4 (1.4%) | 4 (1.5%) | 1.000 |
| Sepsis | 4 | 1 (0.4%) | 8 (3.0%) | 0.127 |
| Vomiting | 4 | 0 (0%) | 1 (0.4%) | 0.818 |
| Infection | 3 | 3 (1.1%) | 5 (1.9%) | 0.818 |
| Thrombocytopenia | 3 | 7 (2.5%) | 0 (0%) | 0.127 |
|  | 4 | 0 (0%) | 1 (0.4%) | 0.818 |
| Hypomagnesemia | 3 | 4 (1.4%) | 2 (0.7%) | 0.974 |
| Myalgia Arthralgia | 3 | 2 (0.7%) | 5 (1.9%) | 0.818 |
| Upper respiratory infection | 3 | 3 (1.1%) | 4 (1.5%) | 0.974 |
| Hypomagnesemia | 4 | 0 (0%) | 1 (0.4%) | 0.818 |
| Lung infection | 3 | 2 (0.7%) | 4 (1.5%) | 0.818 |

*P-values (two-sided) after controlled the false discovery rate from tests on each toxicity and toxicity grades with fisher's exact method.

**Extended Data Table 6 | Detailed Safety and Toxicity Summary**

| | Research (gap schedule) (N=282) | Control (N=269) |
|---|---|---|
| Subjects with an AEs | 282 (100%) | 268 (99.6%) |
| Subjects with an AEs (first 4 cycles) | 282 (100%) | 268 (99.6%) |
| G-CSF not used due to toxicity | 5 (1.8%) | 10 (3.7%) |
| G-CSF not used due to toxicity (first 4 cycles) | 1 (0.4%) | 4 (1.5%) |
| Had transfusion | 145 (51.4%) | 82 (30.5%) |
| Had transfusion (first 4 cycles) | 112 (39.7%) | 57 (21.2%) |
| Missed/Modified doses due to toxicity | 161 (57.1%) | 99 (36.8%) |
| Missed doses due to toxicity | 114 (40.4%) | 76 (28.3%) |
| Modified doses due to toxicity | 109 (38.7%) | 48 (17.8%) |
| Missed/Modified carboplatin due to toxicity | 57 (20.2%) | 26 (9.7%) |
| Missed/Modified olaparib due to toxicity | 77 (27.3%) | 0 (0%) |
| Missed/Modified paclitaxel due to toxicity | 147 (52.1%) | 96 (35.7%) |
| Missed carboplatin due to toxicity | 3 (1.1%) | 4 (1.5%) |
| Modified carboplatin due to toxicity | 55 (19.5%) | 22 (8.2%) |
| Missed olaparib due to toxicity | 7 (2.5%) | - |
| Modified olaparib due to toxicity | 74 (26.2%) | - |
| Missed Paclitaxel due to toxicity | 112 (39.7%) | 74 (27.5%) |
| Modified Paclitaxel due to toxicity | 69 (24.5%) | 46 (17.1%) |
| Participants with AE grade >= 3 | 181 (64.2%) | 158 (58.7%) |
| Participants with AE grade >= 3 (first 4 cycles) | 136 (48.2%) | 96 (35.7%) |
| Participants with SAE | 96 (34.0%) | 93 (34.6%) |
| Participants with SAE (first 4 cycles) | 59 (20.9%) | 49 (18.2%) |
| Participants with SAE related to carboplatin | 60 (21.3%) | 49 (18.2%) |
| Participants with SAE related to paclitaxel | 63 (22.3%) | 47 (17.5%) |
| Participants with SAE related to olaparib | 45 (16.0%) | - |
| Participants with Life-threatening SAE | 9 (3.2%) | 5 (1.9%) |
| Participants with SAE Resulted in death | 1 (0.4%) | 1 (0.4%) |
| Participants discontinued due to toxicity | 21 (7.4%) | 22 (8.2%) |
| Participants died | 32 (11.3%) | 32 (11.9%) |

# Reporting Summary

## Statistics

For all statistical analyses, confirm that the following items are present in the figure legend, table legend, main text, or Methods section.

| n/a | Confirmed | |
|-----|-----------|---|
| ☐ | ☒ | The exact sample size (*n*) for each experimental group/condition, given as a discrete number and unit of measurement |
| ☒ | ☐ | A statement on whether measurements were taken from distinct samples or whether the same sample was measured repeatedly |
| ☐ | ☒ | The statistical test(s) used AND whether they are one- or two-sided *Only common tests should be described solely by name; describe more complex techniques in the Methods section.* |
| ☐ | ☒ | A description of all covariates tested |
| ☐ | ☒ | A description of any assumptions or corrections, such as tests of normality and adjustment for multiple comparisons |
| ☐ | ☒ | A full description of the statistical parameters including central tendency (e.g. means) or other basic estimates (e.g. regression coefficient) AND variation (e.g. standard deviation) or associated estimates of uncertainty (e.g. confidence intervals) |
| ☐ | ☒ | For null hypothesis testing, the test statistic (e.g. *F*, *t*, *r*) with confidence intervals, effect sizes, degrees of freedom and *P* value noted *Give P values as exact values whenever suitable.* |
| ☒ | ☐ | For Bayesian analysis, information on the choice of priors and Markov chain Monte Carlo settings |
| ☐ | ☒ | For hierarchical and complex designs, identification of the appropriate level for tests and full reporting of outcomes |
| ☒ | ☐ | Estimates of effect sizes (e.g. Cohen's *d*, Pearson's *r*), indicating how they were calculated |

*Our web collection on statistics for biologists contains articles on many of the points above.*

## Software and code

Policy information about availability of computer code

| | |
|---|---|
| Data collection | Data were collected with the MACRO database. |
| Data analysis | All statistical analyses were performed with R v4.1.0 (also mentioned in the manuscript within the methods section) |

For manuscripts utilizing custom algorithms or software that are central to the research but not yet described in published literature, software must be made available to editors and reviewers. We strongly encourage code deposition in a community repository (e.g. GitHub). See the Nature Portfolio guidelines for submitting code & software for further information.

## Data

Policy information about availability of data

All manuscripts must include a data availability statement. This statement should provide the following information, where applicable:

- Accession codes, unique identifiers, or web links for publicly available datasets
- A description of any restrictions on data availability
- For clinical datasets or third party data, please ensure that the statement adheres to our policy

Data collected within the PARTNER study will be made available to researchers whose full proposal for their use of the data has been approved by the PARTNER Trial Management Group and whose research includes a clear and comprehensive research plan with statistical considerations adequately completed. The data required for the approved, specified purposes and the trial protocol will be provided, after completion of a data sharing agreement. Data sharing agreements will be set up

by the Trial Steering and Management Groups and will include clear instructions on publication, reporting and usage policy. A minimum dataset of anonymised data will be made available after full publication of the trial and related work. Please address requests for data to ja344@cam.ac.uk

# Research involving human participants, their data, or biological material

Policy information about studies with human participants or human data. See also policy information about sex, gender (identity/presentation), and sexual orientation and race, ethnicity and racism.

| | |
|---|---|
| Reporting on sex and gender | The study focused on breast cancer patients. Although gender or sex identifying information was not collected, it is likely that the majority of participants in this study were assigned female at birth. |
| Reporting on race, ethnicity, or other socially relevant groupings | The study included all participants who met eligibility criteria for the study with no data collected on race, ethnicity, or other socially relevant groupings. |
| Population characteristics | Patients aged between 16 and 70 years with histologically confirmed stage T1-4, N0-2 (tumour or axillary lymph node diameter ≥ 10mm) invasive breast cancer, confirmed ER-negative and HER2-negative, and Eastern Cooperative Oncology Group performance status (ECOG PS) 0-1 were eligible. Detailed eligibility criteria are provided in the methods section. |
| Recruitment | Participants were assessed after consent at 29 UK centres. Eligible participants were randomised to either treatment group(s) or the control. All patients across all sites were assessed for eligibility criteria during their standard clinical evaluation and multidisciplinary team meeting. The trial was offered when it was considered clinically appropriate. There were no self-selection or site-based biases involved. |
| Ethics oversight | The PARTNER trial protocol (NCT03150576) was approved by North West - Haydock Research Ethics Committee (ref: 15/NW/0926) |

Note that full information on the approval of the study protocol must also be provided in the manuscript.

# Field-specific reporting

Please select the one below that is the best fit for your research. If you are not sure, read the appropriate sections before making your selection.

☒ Life sciences  ☐ Behavioural & social sciences  ☐ Ecological, evolutionary & environmental sciences

For a reference copy of the document with all sections, see nature.com/documents/nr-reporting-summary-flat.pdf

# Life sciences study design

All studies must disclose on these points even when the disclosure is negative.

| | |
|---|---|
| Sample size | In this TNBC (gBRCAwt) cohort, a total of 454 patients were needed to attest with 90% power and 5% significance level the null hypothesis of no difference in pCR rate between the two groups versus the alternative of 50% in control group and 65% in research group. Considering a non-compliance of 5%, it was planned to recruit a total of 478 TNBC (gBRCAwt) patients between the control and the selected research group. |
| Data exclusions | The main analysis was conducted based on the modified intention-to-treat (mITT) principle, which included all randomised, eligible patients excluding only those who did not start treatment. The safety analyses included patients who had at least one dose of trial treatment. |
| Replication | The main analyses were performed by another independent statistician and checked against the results of the trial statistician. Two pathologists independently reviewed the slides. The main analysis were performed by another independent statistician and checked against the results of the trial statistician at the final stage of the study. Two pathologist independently reviewed the slides simultaneously. All replications were successful. |
| Randomization | The trial was open label and eligible patients were randomly assigned to either the control group (chemotherapy: paclitaxel 80mg/m2 on day 1, 8 & 15 and carboplatin AUC5 on day 1), or one of the two research groups (chemotherapy with olaparib 150mg twice daily on day -2 to day 10 OR day 3 to day 14) using a minimisation method in a 1:1:1 ratio in Stage 1 and Stage 2 of the trial with a web-based central randomisation system. In Stage 3 (reported here), patients were randomly assigned with a 1:1 ratio to either control or research arm (olaparib 150mg bd on days 3 to day 14). |
| Blinding | This is an open label study. The pathologists were blinded to the treatment arm. |

# Reporting for specific materials, systems and methods

We require information from authors about some types of materials, experimental systems and methods used in many studies. Here, indicate whether each material, system or method listed is relevant to your study. If you are not sure if a list item applies to your research, read the appropriate section before selecting a response.

## Materials & experimental systems

| n/a | Involved in the study |
|---|---|
| ☒ | Antibodies |
| ☒ | Eukaryotic cell lines |
| ☒ | Palaeontology and archaeology |
| ☒ | Animals and other organisms |
| ☐ | ☒ Clinical data |
| ☒ | Dual use research of concern |
| ☒ | Plants |

## Methods

| n/a | Involved in the study |
|---|---|
| ☒ | ChIP-seq |
| ☒ | Flow cytometry |
| ☒ | MRI-based neuroimaging |

## Clinical data

Policy information about clinical studies

All manuscripts should comply with the ICMJE guidelines for publication of clinical research and a completed CONSORT checklist must be included with all submissions.

| | |
|---|---|
| Clinical trial registration | NCT03150576 |
| Study protocol | Available in supplementary materials and uploaded separately as a full protocol. |
| Data collection | Participants were assessed after consent at 29 UK centres between September 2016 to December 2021 |
| Outcomes | The primary endpoint was pathological complete response (pCR), and secondary endpoints included event-free (EFS), and overall survival (OS). |

## Plants

| | |
|---|---|
| Seed stocks | *Report on the source of all seed stocks or other plant material used. If applicable, state the seed stock centre and catalogue number. If plant specimens were collected from the field, describe the collection location, date and sampling procedures.* |
| Novel plant genotypes | *Describe the methods by which all novel plant genotypes were produced. This includes those generated by transgenic approaches, gene editing, chemical/radiation-based mutagenesis and hybridization. For transgenic lines, describe the transformation method, the number of independent lines analyzed and the generation upon which experiments were performed. For gene-edited lines, describe the editor used, the endogenous sequence targeted for editing, the targeting guide RNA sequence (if applicable) and how the editor was applied.* |
| Authentication | *Describe any authentication procedures for each seed stock used or novel genotype generated. Describe any experiments used to assess the effect of a mutation and, where applicable, how potential secondary effects (e.g. second site T-DNA insertions, mosiacism, off-target gene editing) were examined.* |

