## [Peer Review File · Nature]

Manuscript Title: The PARTNER trial of neoadjuvant olaparib with chemotherapy in triple-negative breast cancer

Reviewer Comments & Author Rebuttals

Reviewer Reports on the Initial Version:

Referees' comments:

Referee #1 (Remarks to the Author):

The authors describe the PARTNER trial, in which Olaparib was tested in the neoadjuvant setting in 2 cohorts, of which one cohort were gBRAC1/2 carriers and the other were wildtype gBRCA1/2 TNBC. There were 3 objectives of this study: safety of the combination of drugs, optimal scheduling, and efficacy of chemo plus olaparib in a 48 hour gap-scheduled research arm. Their main finding was that there was no improvement in either pCR rate or estimated EFS/OS at 36 months, from the addition of olaparib. The strength of this study is the detailed characterisation of patients' tumour, the prospectively mapping of g BRCA mutations, that all TNBC were of basal-like phenotype, and the safety assessment. One should congratulate the team for their efforts especially during COVID-19. This is an excellent trial and has clinical impact. Most importantly, standard of care within the UK's National Health Service with recruiting centres adopted carboplatin-based chemotherapy. Whilst the authors elude to translational work, it has not been included in this paper and as such I would suggest that this study would be better placed in NEJM, Lancet or Annals of Oncology to reach the right audience.

Referee #2 (Remarks to the Author):

Authors present results from the Partner study evaluating neoadjuvant carboplatin and paclitaxel +/- olaparib in patients with WT BRCA TNBC. Please consider the following comments:

1. Introduction - Standard of care for TNBC is carboplatin along with pembrolizumab. Mentioning one but not the other is misleading.
2. Authors suggest that in HR deficient cells, PARPi could have activity. Accordingly, inclusion criteria of the trial should have been tumors with HRD. It would be helpful to report % of pts with HRD tumors and activity in that group.
3. Improvement in pCR in TIL high TNBC is not surprising and c/w with multiple previous reports, including meta-analysis (Li et al 2022), which could be cited.
4. While gap scheduling improved tolerability, it is unclear whether the gap scheduling had the same pharmacodynamic effect on HRR/NHEJ as concurrent scheduling, whether it compromised tumor inhibition. Similarly, unclear how much the gap scheduling contributed to pathway inhibition as compared to chemotherapy control. Analysis of on-treatment biopsies or surgical specimens could help address this.
5. Authors mention strength of the study is the strong and comprehensive translational research component, but none of it is reported in the paper. Thus, it might be a general strength of the study

but is not a strength of the manuscript.

Referee #3 (Remarks to the Author):

This is a very well-done work on the addition of Olaparib in the neoadjuvant therapy of TNBC, with interesting results and very important findings for the treatment of early TNBC.

I have only few Improvement suggestions:

- The publication concerns the subgroup of the gBRCAwt cohort. The subgroup of the gBRCA is published elsewhere - has this publication already been made? If yes, please provide the corresponding reference both in the introduction and discussion. If these results have not yet been published, please mention this and indicate that they will be published soon.

- The statistical analysis is correct with all OR and HR provided. I have no suggestions for improvement here. All relevant tables are attached, making the analysis easy to follow.

-For safety and toxicity, I recommend adding percentage values in the results section.

- In the introduction, you mention HRD. Unfortunately, this is not mentioned in the analysis and discussion anymore. Have you collected data on this? Please discuss the HRD analysis briefly.

- In line 176, the word "Benefit" appears twice; please correct this sentence.

- Both the summary and the references are adequate. All relevant studies related to the topic are mentioned.

- The abstract and the summary are appropriate and the main results are summarized.

Congratulation for this important work!

Author Rebuttals to Initial Comments:

Response to Reviewers

Referee #1 (Remarks to the Author):

The authors describe the PARTNER trial, in which Olaparib was tested in the neoadjuvant setting in 2 cohorts, of which one cohort were gBRCA1/2 carriers and the other were wildtype gBRCA1/2 TNBC. There were 3 objectives of this study: safety of the combination of drugs, optimal scheduling, and efficacy of chemo plus olaparib in a 48 hour gap-scheduled research arm. Their main finding was that there was no improvement in either pCR rate or estimated EFS/OS at 36 months, from the addition of olaparib. The strength of this study is the detailed characterisation of patients' tumour, the prospectively mapping of g BRCA mutations, that all TNBC were of basal-like phenotype, and the safety assessment. One should congratulate the team for their efforts especially during COVID-19. This is an excellent trial and has clinical impact. Most importantly, standard of care within the UK's National Health Service with recruiting centres adopted carboplatin-based chemotherapy.

We thank the reviewer for their time, effort and very positive review.

Whilst the authors elude to translational work, it has not been included in this paper and as such I would suggest that this study would be better placed in NEJM, Lancet or Annals of Oncology to reach the right audience.

Whilst we understand the reviewers perspective, the PARTNER TNBC manuscript has a sister manuscript focused on the PARTNER gBRCAm cohort. That manuscript has core pre-clinical work that defined the trial design and subsequent lab work that has helped to explain the outcome results. Undoubtedly the editors will have considered the fit of the current manuscript under review and any other submitted manuscripts. Nature has a broad reach and we believe that this trial and the outcomes of both cohorts will be of general interest given how common breast cancer is and that the outcomes of this trial will reach the appropriate audience and beyond.

The translational work eluded to is of two types neither of which has been published yet:

1. This paper details the complete description of the clinical trial, and as such completes the full clinical results with the parallel publication for the gBRCA mutated group. This latter paper includes detailed laboratory experiments which confirm the clinical findings and provide biological explanations for the extra-ordinary activity of the gap schedule combination of olaparib and carboplatin / paclitaxel in the gBRCA population. In addition the patients with TNBC who are gBRCAwt reported here, who derive no additional benefit from olaparib, do however preserve a standard correlation between achieving a pCR and 3-year EFS and OS.

2. Future translational research on the TNBC gBRCAwt group, will inform us as to whether there are sub-groups which will benefit from olaparib in the same way that the gBRCAm patients have.

We thank the reviewer for highlighting the following strengths:

1. That the study whilst being carried out, changed standard clinical practice within the NHS by adopting carboplatin alongside standard taxane and anthracycline neoadjuvant chemotherapy.
2. That central testing of tumour biopsies allowed us to restrict entry to the study to IHC identified basal-like TNBC. This removes about 20% of the TNBC population who have tumours which behave very differently and are not generally sensitive to chemotherapy in the same way.

Referee #2 (Remarks to the Author):

Authors present results from the Partner study evaluating neoadjuvant carboplatin and paclitaxel +/- olaparib in patients with WT BRCA TNBC. Please consider the following comments:

1. Introduction - Standard of care for TNBC is carboplatin along with pembrolizumab. Mentioning one but not the other is misleading.

In the Introduction we are setting the scene as it was when the study was being discussed and started (2012 to 2016). This is often a challenging thing to do because many things were changing as the results of trials as they were published. In fact, at the time the trial started, carboplatin had not been introduced as standard of care for TNBC and many key opinion leaders were arguing against the routine use of carboplatin in the neoadjuvant setting. However, it seemed to us that this was the direction of travel and we included carboplatin in both arms of the trial. Studies such as the Brightness study were published subsequently demonstrating the benefit of the addition of carboplatin to anthracycline / taxane-based treatment.

KEYNOTE 522 the neoadjuvant trial which added pembrolizumab to carboplatin / paclitaxel followed by anthracycline based chemotherapy is very extensively discussed in the Discussion section, and in our opinion that is the most appropriate place for it.

2. Authors suggest that in HR deficient cells, PARPi could have activity. Accordingly, inclusion criteria of the trial should have been tumors with HRD. It would be helpful to report % of pts with HRD tumors and activity in that group.

Our translational work is ongoing and we do not have any analysis yet of the response of HR deficient tumours to the Olaparib carboplatin / combination. We have however collected ~300+ patient samples for whole genome and transcriptome sequencing and other molecular profiling from the PARTNER trial. This will help us to understand if there were any sub-groups of TNBC more responsive to the addition of olaparib. We agree with the reviewer that knowledge of HRD status is important and this will be dealt with in detail in a separate publication.

We had selected our tumours to be only those classified as basal-like on IHC. These IHC markers were all done by central testing in Cambridge and were reported by our study pathologists. We thereby excluded the 20% or so of TNBC, that do not have a basal-like phenotype.

3. Improvement in pCR in TIL high TNBC is not surprising and c/w with multiple previous reports, including meta-analysis (Li et al 2022), which could be cited.

This meta-analysis was already cited in the manuscript at the end of the discussion.

4. While gap scheduling improved tolerability, it is unclear whether the gap scheduling had the same pharmacodynamic effect on HRR/NHEJ as concurrent scheduling, whether it compromised tumor inhibition. Similarly, unclear how much the gap scheduling contributed to pathway inhibition as compared to chemotherapy control. Analysis of on-treatment biopsies or surgical specimens could help address this.

This reviewer makes an excellent point about whether the improved tolerability seen with gap scheduling, would also compromise activity when compared with control. The companion paper reporting the clinical findings in the gBRCAm cohort show clearly that the anti-tumour activity is enhanced and the accompanying laboratory studies define a biological mechanism to explain this synergy. The results for the gap schedule are significantly better than for the control chemotherapy in the gBRCAm cohort. However, for the TNBC population that same definitive effect is not seen and this is discussed in the sister gBRCAm manuscript.

5. Authors mention strength of the study is the strong and comprehensive translational research component, but none of it is reported in the paper. Thus, it might be a general strength of the study but is not a strength of the manuscript.

The wording of this section has been changed to:

The translational research that is ongoing will be published later and should add significantly to the knowledge base for TNBC (gBRCAwt) and accelerate the application of precision medicine and personalised treatment in this group of patients.

Referee #3 (Remarks to the Author):

This is a very well-done work on the addition of Olaparib in the neoadjuvant therapy of TNBC, with interesting results and very important findings for the treatment of early TNBC.

We thank the reviewer for these kind words.

I have only few Improvement suggestions:

- The publication concerns the subgroup of the gBRCAwt cohort. The subgroup of the gBRCA is published elsewhere - has this publication already been made? If yes, please provide the corresponding reference both in the introduction and discussion. If these results have not yet been published, please mention this and indicate that they will be published soon.

The publication of the gBRCAm cohort has just been submitted to Nature. Both cohorts from the PARTNER trial have been accepted by AACR (San Diego) as Oral presentations in the Clinical Trial Plenary Session, on April 8th 2024.

- The statistical analysis is correct with all OR and HR provided. I have no suggestions for improvement here. All relevant tables are attached, making the analysis easy to follow.

Thank you.

-For safety and toxicity, I recommend adding percentage values in the results section.

We have added percentage values in the results section which now reads as follows:

*The research group experienced slightly more grade ≥ 3 AEs than the control group (64.2% versus 58.7%) (**Table 2**). The serious adverse events (SAEs) related to carboplatin were slightly higher in the research group (60 [21.3%]) than the control group (49 [18.2%]) and SAEs related to paclitaxel were also slightly higher in the research group (63 [22.3%]) than the control group (47 [17.5%]). A total of 45 (16.0%) patients in the research group had an SAE related to olaparib. During the whole treatment period, more patients (145 [51.4%]) in the research group had a transfusion compared with patients in the control group (82 [30.5%]). A summary of the worst AE grade ≥ 3 experienced per patient in at least 1% of patients is shown in **Table S5**, and the only AE which is significantly worse in the research group compared with the control group is neutropenia without associated fever (research group 95 [33.7%]; control group 50 [18.6%], $p=0.002$). More patients in the research group than the control group had a missed/modified dose of carboplatin (20.6% versus 9.7% respectively), or a mixed/modified dose of paclitaxel (64.2% versus 44.6% respectively) (see **Table S6** for full details).*

- In the introduction, you mention HRD. Unfortunately, this is not mentioned in the analysis and discussion anymore. Have you collected data on this? Please discuss the HRD analysis briefly.

Our translational work is ongoing and we do not have any analysis yet of the response of HR deficient tumours to the olaparib carboplatin / combination. We have however collected ~300+ patient samples for whole genome and transcriptome sequencing and other molecular profiling from the PARTNER trial. This will help us to understand if there were any sub-groups of TNBC more responsive to the addition of olaparib. We agree with the reviewer that knowledge of HRD status is important and this will be dealt with in detail in a separate publication.

- In line 176, the word "Benefit" appears twice; please correct this sentence.

Corrected

- Both the summary and the references are adequate. All relevant studies related to the topic are mentioned.

- The abstract and the summary are appropriate and the main results are summarized.

Congratulation for this important work!

We very much appreciate these comments, thank you!